# AN OPTIMAL TRANSPORT PERSPECTIVE ON UNPAIRED IMAGE SUPER-RESOLUTION

## ABSTRACT

Real-world image super-resolution (SR) tasks often do not have paired datasets, which limits the application of supervised techniques. As a result, the tasks are usually approached by *unpaired* techniques based on Generative Adversarial Networks (GANs), which yield complex training losses with several regularization terms, e.g., content or identity losses. We theoretically investigate optimization problems which arise in such models and find two surprizing observations. First, the learned SR map is always an *optimal transport* (OT) map. Second, we theoretically prove and empirically show that the learned map is *biased*, i.e., it does not actually transform the distribution of low-resolution images to high-resolution ones. Inspired by these findings, we propose an algorithm for unpaired SR which learns an *unbiased* OT map for the perceptual transport cost. Unlike the existing GAN-based alternatives, our algorithm has a simple optimization objective reducing the need for complex hyperparameter selection and an application of additional regularizations. At the same time, it provides a nearly state-of-the-art performance on the large-scale unpaired AIM19 dataset.

## 1 INTRODUCTION

The problem of image super-resolution (SR) is to reconstruct a high-resolution (HR) image from its low-resolution (LR) counterpart. In many modern deep learning approaches, SR networks are trained in a supervised manner by using synthetic datasets containing LR-HR *pairs* (Lim et al., 2017, §4.1); (Zhang et al., 2018b, §4.1). For example, it is common to create LR images from HR with a simple downscaling, e.g., bicubic (Ledig et al., 2017, §3.2). However, such an artificial setup barely represents the practical setting, in which the degradation is more sophisticated and unknown (Maeda, 2020). This obstacle suggests the necessity of developing methods capable of learning SR maps from *unpaired* data without considering prescribed degradations.

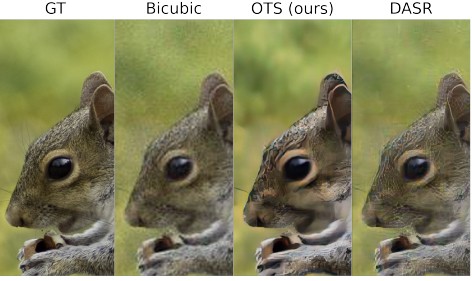

Figure 1: Super-resolution of a squirrel using Bicubic upsample, OTS (ours) and DASR (Wei et al., 2021) methods (4×4 upsample, 370×800 crops).

**Contributions.** We study the unpaired image SR task and its solutions based on Generative Adversarial Networks (Goodfellow et al., 2014, GANs) and analyse them from the Optimal Transport (OT, see (Villani, 2008)) perspective.

1. We investigate the GAN optimization objectives regularized with content losses, which are common in unpaired image SR methods (§5, §4). We prove that the solution to such objectives is always an optimal transport map. We theoretically and empirically show that such maps are *biased* (§7.1), i.e., they do not transform the LR image distribution to the true HR image distribution.

2. We provide an algorithm to fit an unbiased OT map for perceptual transport cost (§6.1) and apply it to the unpaired image SR problem (§7.2). We establish connections between our algorithm and regularized GANs using integral probability metrics (IPMs) as a loss (§6.2).

Our algorithm solves a minimax optimization objective and does not require extensive hyperparameter search, which makes it different from the existing methods for unpaired image SR. At the same time, the algorithm provides a nearly state-of-art performance in the unpaired image SR problem (§7.2).

**Notation.** We use $\mathcal{X}, \mathcal{Y}$ to denote Polish spaces and $\mathcal{P}(\mathcal{X}), \mathcal{P}(\mathcal{Y})$ to denote the respective sets of probability distributions on them. We denote by $\Pi(\mathbb{P}, \mathbb{Q})$ the set of probability distributions on $\mathcal{X} \times \mathcal{Y}$ with marginals $\mathbb{P}$ and $\mathbb{Q}$. For a measurable map $T : \mathcal{X} \to \mathcal{Y}$, we denote the associated push-forward operator by $T_{\#}$. The expression $\| \cdot \|$ denotes the usual Euclidean norm if not stated otherwise. We denote the space of $\mathbb{Q}$-integrable functions on $\mathcal{Y}$ by $L^1(\mathbb{Q})$.

## 2 UNPAIRED IMAGE SUPER-RESOLUTION TASK

In this section, we formalize the *unpaired* image super-resolution task that we consider (Figure 2).

Let $\mathbb{P}$ and $\mathbb{Q}$ be two distributions of LR and HR images, respectively, on spaces $\mathcal{X}$ and $\mathcal{Y}$, respectively. We assume that $\mathbb{P}$ is obtained from $\mathbb{Q}$ via some *unknown* degradation. The learner has access to unpaired random samples from $\mathbb{P}$ and $\mathbb{Q}$. The task is to fit a map $T : \mathcal{X} \to \mathcal{Y}$ satisfying $T_{\#}\mathbb{P} = \mathbb{Q}$ which *inverts* the degradation.

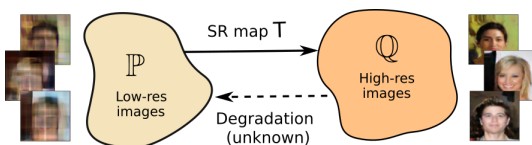

Figure 2: The task of super-resolution we consider.

We highlight that the image SR task is theoretically ill-posed for two reasons.

1. **Non-existence.** The degradation filter may be *non-injective* and, consequently, *non-invertible*. This is a theoretical obstacle to learn one-to-one SR maps $T$.

2. **Ambiguity.** There might exist *multiple* maps satisfying $T_{\#}\mathbb{P} = \mathbb{Q}$ but only one inverting the degradation. With no prior knowledge about the correspondence between $\mathbb{P}$ and $\mathbb{Q}$, it is unclear how to pick this particular map.

**The first issue** is usually not taken into account in practice. Most existing paired and unpaired SR methods learn one-to-one SR maps $T$, see (Ledig et al., 2017; Lai et al., 2017; Wei et al., 2021).

**The second issue** is typically softened by regularizing the model with the content loss. In the real-world, it is reasonable to assume that HR and the corresponding LR images are close. Thus, the fitted SR map $T$ is expected to only *slightly* change the input image. Formally, one may require the learned map $T$ to have the small value of

$$\mathcal{R}_c(T) \stackrel{def}{=} \int_{\mathcal{Y}} c\big(x, T(x)\big)d\mathbb{P}(x), \tag{1}$$

where $c : \mathcal{X} \times \mathcal{Y} \to \mathbb{R}_+$ is a function estimating how different the inputs are. The most popular example is the $\ell^1$ *identity* loss, i.e, formulation (1) for $\mathcal{X} = \mathcal{Y} = \mathbb{R}^D$ and $c(x, y) = \|x - y\|_1$.

More broadly, losses $\mathcal{R}_c(T)$ are typically called *content* losses and incorporated into training objectives of methods for SR (Lugmayr et al., 2019a, §3.4), (Kim et al., 2020, §3) and other unpaired tasks beside SR (Taigman et al., 2016, §4), (Zhu et al., 2017, §5.2) as regularizers. They stimulate the learned map $T$ to minimally change the image content.

## 3 BACKGROUND ON OPTIMAL TRANSPORT

In this section, we give the key concepts of the OT theory (Villani, 2008) that we use in our paper.

**Primal form**. For two distributions $\mathbb{P} \in \mathcal{P}(\mathcal{X})$ and $\mathbb{Q} \in \mathcal{P}(\mathcal{Y})$ and a transport cost $c : \mathcal{X} \times \mathcal{Y} \to \mathbb{R}$, Monge's primal formulation of the *optimal transport cost* is as follows:

$$\text{Cost}(\mathbb{P}, \mathbb{Q}) \stackrel{def}{=} \inf_{T_{\#}\mathbb{P}=\mathbb{Q}} \int_{\mathcal{X}} c\big(x, T(x)\big)d\mathbb{P}(x), \tag{2}$$

where the minimum is taken over the measurable functions (transport maps) $T : \mathcal{X} \to \mathcal{Y}$ that map $\mathbb{P}$ to $\mathbb{Q}$, see Figure 3a. The optimal $T^*$ is called the *optimal transport map*.

Note that (2) is not symmetric, and this formulation does not allow mass splitting, i.e., for some $\mathbb{P}, \mathbb{Q}$ there may be no map $T$ that satisfies $T_{\#}\mathbb{P} = \mathbb{Q}$. Thus, (Kantorovitch, 1958) proposed the relaxation:

$$\text{Cost}(\mathbb{P}, \mathbb{Q}) \stackrel{def}{=} \inf_{\pi \in \Pi(\mathbb{P}, \mathbb{Q})} \int_{\mathcal{X} \times \mathcal{Y}} c(x, y)d\pi(x, y), \tag{3}$$

where the minimum is taken over the transport plans $\pi$, i.e., the measures on $\mathcal{X} \times \mathcal{Y}$ whose marginals are $\mathbb{P}$ and $\mathbb{Q}$ (Figure 3b). The optimal $\pi^* \in \Pi(\mathbb{P}, \mathbb{Q})$ is called the *optimal transport plan*.

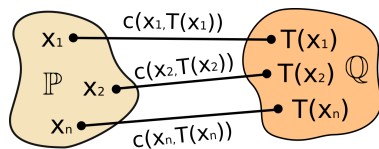 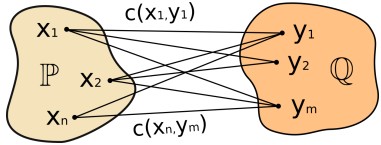

(a) Monge's formulation of OT.      (b) Kantorovich's formulation of OT.

Figure 3: Monge's and Kantorovich's formulations of Optimal Transport.

With mild assumptions on the transport cost $c(x, y)$ and distributions $\mathbb{P}$, $\mathbb{Q}$, the minimizer $\pi^*$ of (3) always exists (Villani, 2008, Theorem 4.1) but might not be unique. If $\pi^*$ is of the form $[\text{id}, T^*]_\#\mathbb{P} \in \Pi(\mathbb{P}, \mathbb{Q})$ for some $T^*$, then $T^*$ is an optimal transport map that minimizes (2).

**Dual form**. The dual form (Villani, 2003) of OT cost (3) is as follows:

$$\text{Cost}(\mathbb{P}, \mathbb{Q}) = \sup_f \left[ \int_{\mathcal{X}} f^c(x) d\mathbb{P}(x) + \int_{\mathcal{Y}} f(y) d\mathbb{Q}(y) \right]; \qquad (4)$$

here $\sup$ is taken over all $f \in \mathcal{L}^1(\mathbb{Q})$, and $f^c(x) = \inf_{y \in \mathcal{Y}} \left[ c(x, y) - f(y) \right]$ is the $c$-transform of $f$.

## 4 RELATED WORK

UNPAIRED IMAGE SUPER-RESOLUTION. Existing approaches to unpaired image SR mainly solve the problem in two steps. One group of approaches learn the degradation operation at the first step and then train a super-resolution model in a supervised manner using generated pseudo-pairs, see (Bulat et al., 2018; Fritsche et al., 2019). Another group of approaches (Yuan et al., 2018; Maeda, 2020) firstly learn a mapping from real-world LR images to "clean" LR images, i.e., HR images, downscaled using predetermined (e.g., bicubic) operation, and then a mapping from "clean" LR to HR images. Most methods are based on CycleGAN (Zhu et al., 2017), initially designed for the domain transfer task, and utilize cycle-consistency loss. Methods are also usually endowed with several other losses, e.g. content (Kim et al., 2020, §3), identity (Wang et al., 2021, §3.2) or perceptual (Lugmayr et al., 2019a, §3.4).

OPTIMAL TRANSPORT IN GENERATIVE MODELS. The majority of existing OT-based generative models employ OT cost as the loss function to update the generative network, e.g., see (Arjovsky et al., 2017). These methods are out of scope of the present paper, since they do not compute OT maps. Existing methods to compute the OT map approach the primal (2), (3) or dual form (4). Primal-form methods (Lu et al., 2020; Xie et al., 2019; Bousquet et al., 2017; Balaji et al., 2020) optimize complex GAN objectives such as (5) and provide biased solutions (§5, §7.1). For a comprehensive overview of dual-form methods, we refer to (Korotin et al., 2021). The authors conduct an evaluation of OT methods for the quadratic cost $c(x, y) = \|x - y\|^2$. According to them, the best performing method is $\lfloor \text{MM:R} \rceil$. It is based on the variational reformulation of (4), which is a particular case of our formulation (12). Extensions of $\lfloor \text{MM:R} \rceil$ appear in (Rout et al., 2022; Fan et al., 2021).

## 5 BIASED OPTIMAL TRANSPORT IN GANs

In this section, we establish connections between GAN methods regularized by content losses (1) and OT. Such GANs are popular in a variety of tasks beside SR, e.g., style transfer (Huang et al., 2018). The theoretical analysis in this section holds for these tasks as well. However, since we empirically demonstrate the findings on a practically important SR problem, we keep the corresponding notation throughout §5. A common approach to solve the unpaired SR via GANs

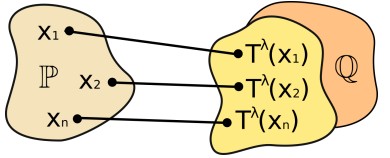

Figure 4: Illustration of Lemma 1. The solution $T^\lambda$ of (5) is an OT map from $\mathbb{P}$ to $T^\lambda_\#\mathbb{P}$. In general, $T^\lambda_\#\mathbb{P} \neq \mathbb{Q}$ (Thm. 1).

is to define a loss function $\mathcal{D} : \mathcal{P}(\mathcal{Y}) \times \mathcal{P}(\mathcal{Y}) \to \mathbb{R}_+$ and train a generative neural network $T$ via minimizing

$$\inf_{T: \mathcal{X} \mapsto \mathcal{Y}} \left[ \mathcal{D}(T_\#\mathbb{P}, \mathbb{Q}) + \lambda \mathcal{R}_c(T) \right]. \qquad (5)$$

The term $\mathcal{D}(T_\#\mathbb{P}, \mathbb{Q})$ ensures that the generated distribution $T_\#\mathbb{P}$ of SR images is close to the true HR distribution $\mathbb{Q}$; the second term $\mathcal{R}_c(T)$ is the content loss (1). For convenience, we assume that $\mathcal{D}(\mathbb{Q}, \mathbb{Q}) = 0$ for all $\mathbb{Q} \in \mathcal{P}(\mathcal{Y})$. Two most popular examples of $\mathcal{D}$ are the Jensen–Shannon divergence (Goodfellow et al., 2014), i.e., the vanilla GAN loss, and the Wasserstein-1 loss (Arjovsky & Bottou, 2017). In unpaired SR methods, the optimization objectives are typically more complex than (5). In addition to the content or identity loss (1), several other regularizations are usually introduced, see §4. In Appendix F, we show that the learning objectives of popular SR methods can be represented as (5).

For a theoretical analysis, we stick to the basic formulation regularized with generic content loss (5). It represents the simplest and straightforward SR setup. We prove the following lemma, which connects the solution $T^\lambda$ of (5) and optimal maps for transport cost $c(x, y)$.

**Lemma 1** (The solution of the regularized GAN is an OT map). *Assume that $\lambda > 0$ and the minimizer $T^\lambda$ of (5) exists. Then $T^\lambda$ is an OT map between $\mathbb{P}$ and $\mathbb{Q}^\lambda \stackrel{\text{def}}{=} T^\lambda_\# \mathbb{P}$ for cost $c(x, y)$, i.e., it minimizes*

$$\inf_{T_\# \mathbb{P} = \mathbb{Q}^\lambda} \mathcal{R}_c(T) = \inf_{T_\# \mathbb{P} = \mathbb{Q}^\lambda} \int_{\mathcal{X}} c\big(x, T(x)\big) d\mathbb{P}(x).$$

*Proof.* Assume that $T^\lambda$ is not an optimal map between $\mathbb{P}$ and $T^\lambda_\# \mathbb{P}$. Then there exists a more optimal $T^\dagger$ satisfying $T^\dagger_\# \mathbb{P} = T^\lambda_\# \mathbb{P}$ and $\mathcal{R}_c(T^\dagger) < \mathcal{R}_c(T^\lambda)$. We substitute this $T^\dagger$ to (5) and derive

$$\mathcal{D}(T^\dagger_\# \mathbb{P}, \mathbb{Q}) + \lambda \mathcal{R}_c(T^\dagger) = \mathcal{D}(T^\lambda_\# \mathbb{P}, \mathbb{Q}) + \lambda \mathcal{R}_c(T^\dagger) < \mathcal{D}(T^\lambda_\# \mathbb{P}, \mathbb{Q}) + \lambda \mathcal{R}_c(T^\lambda),$$

which is a contradiction, since $T^\lambda$ is a minimizer of (5), but $T^\dagger$ provides the smaller value. $\qquad\square$

Our Lemma 1 states that the minimizer $T^\lambda$ of a regularized GAN problem is *always* an OT map between $\mathbb{P}$ and the distribution $\mathbb{Q}^\lambda$ generated by the same $T^\lambda$ from $\mathbb{P}$. However, below we prove that $\mathbb{Q}^\lambda \neq \mathbb{Q}$, i.e., $T^\lambda$ **does not** actually produce the distribution of HR images (Figure 4). To begin with, we state and prove the following auxiliary result.

**Lemma 2** (Reformulation of the regularized GAN via distributions). *Under the assumptions of Lemma 1, let $\mathcal{X} = \mathcal{Y}$ be a compact subset of $\mathbb{R}^D$ with negligible boundary. Let $\mathbb{P} \in \mathcal{P}(\mathcal{X})$ be absolutely continuous, $\mathbb{Q} \in \mathcal{P}(\mathcal{Y})$ and $c(x, y) = \|x - y\|^p$ with $p > 1$. Then (5) is equivalent to*

$$\inf_{\mathbb{Q}' \in \mathcal{P}(\mathcal{Y})} \mathcal{F}(\mathbb{Q}') \stackrel{\text{def}}{=} \inf_{\mathbb{Q}' \in \mathcal{P}(\mathcal{Y})} \big[ \mathcal{D}(\mathbb{Q}', \mathbb{Q}) + \lambda \cdot \text{Cost}(\mathbb{P}, \mathbb{Q}') \big], \qquad (6)$$

*and the solutions of (5) and (6) are related as $\mathbb{Q}^\lambda = T^\lambda_\# \mathbb{P}$, where $\mathbb{Q}^\lambda$ is the minimizer of (6).*

*Proof.* We derive

$$\inf_{T:\mathcal{X} \mapsto \mathcal{Y}} \big[ \mathcal{D}(T_\# \mathbb{P}, \mathbb{Q}) + \lambda \mathcal{R}_c(T) \big] = \inf_{T:\mathcal{X} \mapsto \mathcal{Y}} \Big[ \mathcal{D}(T_\# \mathbb{P}, \mathbb{Q}) + \lambda \int_{\mathcal{X}} c\big(x, T(x)\big) d\mathbb{P}(x) \Big] = \qquad (7)$$

$$\inf_{T:\mathcal{X} \mapsto \mathcal{Y}} \big[ \mathcal{D}(T_\# \mathbb{P}, \mathbb{Q}) + \lambda \cdot \text{Cost}(\mathbb{P}, T_\# \mathbb{P}) \big] = \inf_{\mathbb{Q}' \in \mathcal{P}(\mathcal{Y})} \big[ \mathcal{D}(\mathbb{Q}', \mathbb{Q}) + \lambda \cdot \text{Cost}(\mathbb{P}, \mathbb{Q}') \big]. \qquad (8)$$

In transition from (7) to (8), we use the definition of OT cost (2) and our Lemma 1, which states that the minimizer $T^\lambda$ of (5) is an OT map, i.e., $\int_{\mathcal{X}} c\big(x, T^\lambda(x)\big) d\mathbb{P}(x) = \text{Cost}(\mathbb{P}, T^\lambda_\# \mathbb{P})$. The equality in (8) follows from the fact that $\mathbb{P}$ is abs. cont. and $c(x, y) = \|x - y\|^p$: for all $\mathbb{Q}' \in \mathcal{P}(\mathcal{Y})$ there exists a (unique) solution $T$ to the Monge OT problem (2) for $\mathbb{P}, \mathbb{Q}'$ (Santambrogio, 2015, Thm. 1.17). $\qquad\square$

In the following Theorem, we prove that, in general, $\mathbb{Q}^\lambda \neq \mathbb{Q}$ for the minimizer $\mathbb{Q}^\lambda$ of (6).

**Theorem 1** (The distribution solving the regularized GAN problem is always biased). *Under the assumptions of Lemma 2, assume that the first variation (Santambrogio, 2015, Definition 7.12) of the functional $\mathbb{Q}' \mapsto \mathcal{D}(\mathbb{Q}', \mathbb{Q})$ at the point $\mathbb{Q}' = \mathbb{Q}$ exists and is equal to zero. This means that $\mathcal{D}(\mathbb{Q} + \epsilon \Delta \mathbb{Q}) = \mathcal{D}(\mathbb{Q}, \mathbb{Q}) + o(\epsilon)$ for every signed measure $\Delta \mathbb{Q}$ of zero total mass and $\epsilon \geq 0$ such that $\mathbb{Q} + \epsilon \Delta \mathbb{Q} \in \mathcal{P}(\mathcal{Y})$. Then, if $\mathbb{P} \neq \mathbb{Q}$, then $\mathbb{Q}' = \mathbb{Q}$ does not deliver the minimum to $\mathcal{F}$.*

Before proving Theorem 1, we highlight that the assumption about the vanishing first variation of $\mathbb{Q}' \mapsto \mathcal{D}(\mathbb{Q}', \mathbb{Q})$ at $\mathbb{Q}' = \mathbb{Q}$ is *reasonable*. In Appendix A, we prove that this assumption holds for the popular GAN discrepancies $\mathcal{D}(\mathbb{Q}', \mathbb{Q})$, e.g., $f$-divergences (Nowozin et al., 2016), Wasserstein distances (Arjovsky et al., 2017), and Maximum Mean Discrepancies (Li et al., 2017).

*Proof.* Let $\Delta \mathbb{Q} = \mathbb{P} - \mathbb{Q}$ denote the difference measure of $\mathbb{P}$ and $\mathbb{Q}$. It has zero total mass and $\forall \epsilon \in [0, 1]$ it holds that $\mathbb{Q} + \epsilon \Delta \mathbb{Q} = \epsilon \mathbb{P} + (1 - \epsilon) \mathbb{Q}$ is a mixture distribution of probability distributions $\mathbb{P}$ and $\mathbb{Q}$. As a result, for all $\epsilon \in [0, 1]$, we have

$$\mathcal{F}(\mathbb{Q} + \epsilon \Delta \mathbb{Q}) = \mathcal{D}(\mathbb{Q} + \epsilon \Delta \mathbb{Q}, \mathbb{Q}) + \lambda \cdot \text{Cost}(\mathbb{P}, \mathbb{Q} + \epsilon \Delta \mathbb{Q}) =$$

$$\mathcal{D}(\mathbb{Q}, \mathbb{Q}) + o(\epsilon) + \lambda \cdot \text{Cost}(\mathbb{P}, \epsilon \mathbb{P} + (1 - \epsilon) \mathbb{Q}) \leq \qquad (9)$$

$$o(\epsilon) + \lambda \cdot \epsilon \cdot \text{Cost}(\mathbb{P}, \mathbb{P}) + \lambda \cdot (1 - \epsilon) \cdot \text{Cost}(\mathbb{P}, \mathbb{Q}) = o(\epsilon) + \lambda \cdot (1 - \epsilon) \cdot \text{Cost}(\mathbb{P}, \mathbb{Q}) = \qquad (10)$$

$$\underbrace{\lambda \cdot \text{Cost}(\mathbb{P}, \mathbb{Q})}_{=\mathcal{F}(\mathbb{Q})} - \lambda \cdot \epsilon \cdot \underbrace{\text{Cost}(\mathbb{P}, \mathbb{Q})}_{>0} + o(\epsilon),$$

where in transition from (9) to (10), we use $\mathcal{D}(\mathbb{Q}, \mathbb{Q}) = 0$ and exploit the convexity of the OT cost (Villani, 2003, Theorem 4.8). In (10), we use $\text{Cost}(\mathbb{P}, \mathbb{P}) = 0$. We see that $\mathcal{F}(\mathbb{Q} + \epsilon \Delta \mathbb{Q})$ is smaller then $\mathcal{F}(\mathbb{Q})$ for sufficiently small $\epsilon > 0$, i.e., $\mathbb{Q}' = \mathbb{Q}$ does not minimize $\mathcal{F}$. $\qquad\square$

**Corollary 1.** *Under the assumptions of Theorem 1, the solution $T^\lambda$ of regularized GAN (5) is biased, i.e., it does not satisfy $T^\lambda_\# \mathbb{P} = \mathbb{Q}$ and does not transform LR images to true HR ones.*

Additionally, we provide a toy example that further illustrates the issue with the bias.

**Example 1.** *Consider $\mathcal{X} = \mathcal{Y} = \mathbb{R}^1$. Let $\mathbb{P} = \frac{1}{2}\delta_0 + \frac{1}{2}\delta_2$, $\mathbb{Q} = \frac{1}{2}\delta_1 + \frac{1}{2}\delta_3$ be distributions concentrated at $\{0, 2\}$ and $\{1, 3\}$, respectively. Put $c(x, y) = |x - y|$ to be the content loss. Also, let $\mathcal{D}$ to be the OT cost for $|x - y|^2$. Then for $\lambda = 0$ there exist two maps between $\mathbb{P}$ and $\mathbb{Q}$ that deliver the same minimal value for (5), namely $T(0) = 1, T(2) = 3$ and $T(0) = 3, T(2) = 1$. For $\lambda > 0$, the optimal solution of the problem (5) is unique, **biased** and given by $T(0) = 1 - \frac{\lambda}{2}, T(2) = 3 - \frac{\lambda}{2}$.*

*Proof.* Let $T(0) = t_0$ and $T(2) = t_2$. Then $T_\# \mathbb{P} = \frac{1}{2}\delta_{t_0} + \frac{1}{2}\delta_{t_2}$, and now (5) becomes

$$\min_{t_0, t_2} \left[ \min \left\{ \frac{1}{2}(t_0 - 1)^2 + \frac{1}{2}(t_2 - 3)^2; \frac{1}{2}(t_0 - 3)^2 + \frac{1}{2}(t_2 - 1)^2 \right\} + \lambda \left\{ \frac{1}{2}|0 - t_0| + \frac{1}{2}|2 - t_2| \right\} \right],$$

where the second term is $\mathcal{R}_c(T)$ and the first term is the OT cost $\mathcal{D}(T_\# \mathbb{P}, \mathbb{Q})$ expressed as the minimum over the transport costs of two possible transport maps $t_0 \mapsto 1; t_2 \mapsto 3$ and $t_0 \mapsto 3; t_2 \mapsto 1$. The minimizer can be derived analytically and equals $t_0 = 1 - \frac{\lambda}{2}, t_2 = 3 - \frac{\lambda}{2}$. $\quad\square$

In Example 1, $T^\lambda_\# \mathbb{P} = \mathbb{Q}^\lambda$ *never* matches $\mathbb{Q}$ exactly for $\lambda > 0$. In §7.1, we conduct an evaluation of maps obtained via minimizing objective (5) on the synthetic benchmark by (Korotin et al., 2021). We empirically demonstrate that the bias exists and it is indeed a notable practical issue.

**Remarks.** Throughout this section, we enforce additional assumptions on (5), e.g., we restrict our analysis to content losses $c(\cdot, \cdot)$, which are powers of Euclidean norms $\| \cdot \|^p$. This is needed to make the derivations concise and to be able to exploit the available results in OT. We think that the provided results hold under more general assumptions and leave this question open for future studies.

## 6 UNBIASED OPTIMAL TRANSPORT SOLVER

In §6.1, we derive our algorithm to compute OT maps. Importantly, in §6.2, we detail its differences and similarities with regularized GANs, which we discussed in §5. Our algorithm is suitable for general costs and generalizes the OT algorithm for the quadratic cost by (Rout et al., 2022).

### 6.1 MINIMAX OPTIMIZATION ALGORITHM

We derive a minimax optimization problem to recover the optimal transport map from $\mathbb{P}$ to $\mathbb{Q}$. We expand the dual form (4). To do this, we first note that

$$\int_{\mathcal{X}} f^c(x) d\mathbb{P}(x) = \int_{\mathcal{X}} \inf_{y \in \mathcal{Y}} \left\{ c(x, y) - f(y) \right\} d\mathbb{P}(x) = \inf_{T: \mathcal{X} \to \mathcal{Y}} \int_{\mathcal{X}} \left\{ c(x, T(x)) - f(T(x)) \right\} d\mathbb{P}(x). \quad (11)$$

Here we replace the optimization over points $y \in \mathcal{Y}$ with an equivalent optimization over the functions $T : \mathcal{X} \to \mathcal{Y}$. This is possible due to the Rockafellar interchange theorem (Rockafellar, 1976, Theorem 3A). Substituting (11) to (4), we have

$$\text{Cost}(\mathbb{P}, \mathbb{Q}) = \sup_f \inf_{T: \mathcal{X} \to \mathcal{Y}} \left[ \int_{\mathcal{Y}} f(y) d\mathbb{Q}(y) + \int_{\mathcal{X}} \left\{ c(x, T(x)) - f(T(x)) \right\} d\mathbb{P}(x) \right] \quad (12)$$

We denote the expression under the $\sup \inf$ by $\mathcal{L}(f, T)$. Now we show that by solving the saddle point problem (12) one can obtain the OT map $T^*$.

**Lemma 3** (OT maps solve the saddle point problem). *Assume that the OT map $T^*$ between $\mathbb{P}, \mathbb{Q}$ for cost $c(x, y)$ exists. Then, for every optimal potential $f^* \in \arg\sup_f \left[ \inf_{T: \mathcal{X} \to \mathcal{Y}} \mathcal{L}(f, T) \right]$ of (12),*

$$T^* \in \arg\inf_{T: \mathcal{X} \to \mathcal{Y}} \int_{\mathcal{X}} \left\{ c(x, T(x)) - f(T(x)) \right\} d\mathbb{P}(x). \quad (13)$$

*Proof.* Since $f^*$ is optimal, we have $\inf_{T: \mathcal{X} \to \mathcal{Y}} \mathcal{L}(f^*, T) = \text{Cost}(\mathbb{P}, \mathbb{Q})$. We use $T^*_\# \mathbb{P} = \mathbb{Q}$ and the change of variables $y = T^*(x)$ to derive $\int_{\mathcal{X}} f^*(T^*(x)) d\mathbb{P} = \int_{\mathcal{Y}} f^*(y) d\mathbb{Q}$. Substituting this equality into (12), we obtain $\mathcal{L}(f^*, T^*) = \int_{\mathcal{X}} c(x, T^*(x)) d\mathbb{P}(x) = \text{Cost}(\mathbb{P}, \mathbb{Q})$, i.e., (13) holds. $\quad\square$

Our Lemma 3 states that one can solve a saddle point problem (12) and extract an OT map $T^*$ between $\mathbb{P}, \mathbb{Q}$ from the optimal pair $(f^*, T^*)$. Analogous result but only for the (Q-embedded) quadratic cost can be found in (Rout et al., 2022). For general $\mathbb{P}, \mathbb{Q}$, the $\arg\inf_T$ set for an optimal $f^*$ might contain

---

**Algorithm 1:** OT solver to compute the OT map between $\mathbb{P}$ and $\mathbb{Q}$ for transport cost $c(x,y)$.

---

**Input** : distributions $\mathbb{P}, \mathbb{Q}$ accessible by samples; mapping network $T_\theta : \mathcal{X} \to \mathcal{Y}$;

potential $f_\omega : \mathcal{X} \to \mathbb{R}$; transport cost $c : \mathcal{X} \times \mathcal{Y} \to \mathbb{R}$; number $K_T$ of inner iters;

**Output:** approximate OT map $(T_\theta)_\# \mathbb{P} \approx \mathbb{Q}$;

**repeat**

   Sample batches $X \sim \mathbb{P}, Y \sim \mathbb{Q}$;

   $\mathcal{L}_f \leftarrow \frac{1}{|Y|} \sum_{y \in Y} f_\omega(y) - \frac{1}{|X|} \sum_{x \in X} f_\omega\big(T_\theta(x)\big)$;

   Update $\omega$ by using $\frac{\partial \mathcal{L}_f}{\partial \omega}$ to maximize $\mathcal{L}_f$;

   **for** $k_T = 1, 2, \ldots, K_T$ **do**

      Sample batch $X \sim \mathbb{P}$;

      $\mathcal{L}_T \leftarrow \frac{1}{|X|} \sum_{x \in X} \big[c\big(x, T_\theta(x)\big) - f_\omega\big(T_\theta(x)\big)\big]$;

      Update $\theta$ by using $\frac{\partial \mathcal{L}_T}{\partial \theta}$ to minimize $\mathcal{L}_T$;

**until** *not converged*;

---

not only OT map $T^*$ but other functions as well. However, our experiments (§7) show that this is not a serious issue in practice. To solve the optimization problem (12), we approximate the potential $f$ and map $T$ with neural networks $f_\omega$ and $T_\theta$, respectively. We train the networks with stochastic gradient ascent-descent by using random batches from $\mathbb{P}, \mathbb{Q}$.

The practical optimization procedure is detailed in Algorithm 1. We call this procedure an **Optimal Transport Solver** (OTS).

## 6.2 REGULARIZED GANS VS. OPTIMAL TRANSPORT SOLVER

In this subsection, we discuss similarities and differences between our optimization objective (12) and the objective of regularized GANs (5). We establish an intriguing connection between GANs that use *integral probability metrics* (IPMs) as $\mathcal{D}$. A discrepancy $\mathcal{D} : \mathcal{P}(\mathcal{Y}) \times \mathcal{P}(\mathcal{Y}) \to \mathbb{R}_+$ is an IPM if

$$\mathcal{D}(\mathbb{Q}_1, \mathbb{Q}_2) = \sup_{f \in \mathcal{F}} \Big[ \int_\mathcal{Y} f(y) d\mathbb{Q}_2(y) - \int_\mathcal{Y} f(y) d\mathbb{Q}_1(y) \Big], \tag{14}$$

where the maximization is performed over some certain class $\mathcal{F}$ of functions (discriminators) $f : \mathcal{Y} \to \mathbb{R}$. The most popular example of $\mathcal{D}$ is the Wasserstein-1 loss (Arjovsky & Bottou, 2017), where $\mathcal{F}$ is a class of 1-Lipschitz functions. For other IPMs, see (Mroueh et al., 2017, Table 1).

Substituting (14) to (5) yields the saddle-point optimization problem for the **regularized IPM GAN**:

$$\inf_{T:\mathcal{X} \to \mathcal{Y}} \Big[ \sup_{f \in \mathcal{F}} \Big\{ \int_\mathcal{Y} f(y) d\mathbb{Q}(y) - \int_\mathcal{X} f\big(T(x)\big) d\mathbb{P}(x) \Big\} + \lambda \int_\mathcal{X} c\big(x, T(x)\big) d\mathbb{P}(x) \Big\}\Big]$$

$$= \inf_{T:\mathcal{X} \to \mathcal{Y}} \sup_{f \in \mathcal{F}} \Big[ \int_\mathcal{Y} f(y) d\mathbb{Q}(y) + \int_\mathcal{X} \big\{ \lambda \cdot c\big(x, T(x)\big) - f\big(T(x)\big) \big\} d\mathbb{P}(x) \Big]. \tag{15}$$

We emphasize that the expression inside (15) for $\lambda = 1$ is similar to the expression in OTS optimization (12). Below we highlight the **key differences** between (12) and (15).

**First**, in OTS the map $T$ is a solution to the inner optimization problem, while in IPM GAN the generator $T$ is a solution to the outer problem. Swapping $\inf_T$ and $\sup_f$ is *prohibited* and, in general, yields a different problem, e.g., $1 = \inf_x \sup_y \cos(x+y) \neq \sup_y \inf_x \cos(x+y) = -1$.

**Second**, in OTS the optimization over potential $f$ is unconstrained, while in IPM GAN it must belong to $\mathcal{F}$, some certain restricted class of functions. For example, when $\mathcal{D}$ is the Wasserstein-1 ($\mathbb{W}_1$) IPM, one has to use an additional penalization, e.g., the gradient penalty (Gulrajani et al., 2017). This further complicates the optimization and adds hyperparameters which have to be carefully selected.

**Third**, the optimization of IPM GAN requires selecting a parameter $\lambda$ that balances the content loss $\mathcal{R}_c$ and the discrepancy $\mathcal{D}$. In OTS for all costs $\lambda \cdot c(x,y)$ with $\lambda > 0$, the OT map $T^*$ is the same.

To conclude, **even for $\lambda = 1$, the IPM GAN problem does not match that of OTS.** Table 1 summarizes the differences and the similarities between OTS and regularized IPM GANs.

| | **Optimal Transport Solver (Ours)** | **Regularized IPM GAN** |
|---|---|---|
| Minimax optimization objective | $\sup_f \inf_{T:\mathcal{X}\to\mathcal{Y}} \Big[ \int_{\mathcal{Y}} f(y)d\mathbb{Q}(y) +$ $\int_{\mathcal{X}} \{ c(x,T(x)) - f(T(x)) \} d\mathbb{P}(x) \Big]$ | $\inf_{T:\mathcal{X}\to\mathcal{Y}} \sup_{f\in\mathcal{F}} \Big[ \int_{\mathcal{Y}} f(y)d\mathbb{Q}(y) +$ $\int_{\mathcal{X}} \{ \lambda \cdot c(x,T(x)) - f(T(x)) \} d\mathbb{P}(x) \Big]$ |
| Transport map $T$ (generator) | $T^*$ solves the inner problem (for optimal $f^*$); it is an OT map from $\mathbb{P}$ to $\mathbb{Q}$ (Lemma 3) | $T^*$ solves the outer problem; it is a biased OT map (§5, §7.1) |
| Potential $f$ (discriminator) | Unconstrained $f \in L^1(\mathbb{Q})$ | Constrained $f \in \mathcal{F} \subset L^1(\mathbb{Q})$ A method to impose the constraint is needed. |
| Regularization weight $\lambda$ | N/A | Hyperparameter choice required |

Table 1: Comparison of the optimization objectives of OTS (ours) and regularized IPM GAN.

# 7 EVALUATION

In §7.1, we assess the bias of regularized IPM GANs by using the Wasserstein-2 benchmark (Korotin et al., 2021). In §7.2, we evaluate our method on the large-scale unpaired AIM-19 dataset from (Lugmayr et al., 2019b). In Appendix D, we test it on the CelebA dataset (Liu et al., 2015). The code is written in PyTorch. We list the hyperparameters for Algorithm 1 in Table 4 of Appendix C.

**Neural network architectures.** We use WGAN-QC's (Liu et al., 2019) ResNet (He et al., 2016) architecture for the potential $f_\omega$. In §7.1, where input and output images have the same size, we use UNet[1] (Ronneberger et al., 2015) as a transport map $T_\theta$. In §7.2, the LR input images are $4 \times 4$ times smaller than HR, so we use EDSR network (Lim et al., 2017).

**Transport costs.** In §7.1, we use the *mean squared error* (MSE), i.e., $c(x,y) = \frac{\|x-y\|^2}{\dim(\mathcal{Y})}$. It is equivalent to the quadratic cost but is more convenient due to the normalization. In §7.2, we consider $c(x,y) = b(\text{Up}(x), y)$, where $b$ is a cost between the bicubically upsampled LR image $x^{\text{up}} = \text{Up}(x)$ and HR image $y$. We test $b$ defined as MSE and the *perceptual cost* using features of a pre-trained VGG-16 network (Simonyan & Zisserman, 2014), see Appendix C for details.

## 7.1 ASSESSING THE BIAS IN REGULARIZED GANS

In this section, we empirically confirm the insight of §5 that the solution $T^\lambda$ of (5) may not satisfy $T^\lambda_\# \mathbb{P} = \mathbb{Q}$. Note if $T^\lambda_\# \mathbb{P} = \mathbb{Q}$, then by our Lemma 1, we conclude that $T^\lambda \equiv T^*$, where $T^*$ is an OT map from $\mathbb{P}$ to $\mathbb{Q}$ for $c(x,y)$. Thus, to access the bias, it is reasonable to compare the learned map $T^\lambda$ with the ground truth OT map $T^*$ for $\mathbb{P}, \mathbb{Q}$.

For evaluation, we use the Wasserstein-2 benchmark (Korotin et al., 2021). It provides high-dimensional continuous pairs $\mathbb{P}, \mathbb{Q}$ with an *analytically known* OT map $T^*$ for the quadratic cost $c(x,y) = \|x-y\|^2$. We use their "Early" images benchmark pair. It simulates the image deblurring setup, i.e., $\mathcal{X} = \mathcal{Y}$ is the space of $64 \times 64$ RGB images, $\mathbb{P}$ is blurry faces, $\mathbb{Q}$ is clean faces satisfying $\mathbb{Q} = T^*_\# \mathbb{P}$, where $T^*$ is an analytically known OT map, see the 1st and 2nd lines in Figure 5.

To quantify the learned maps from $\mathbb{P}$ to $\mathbb{Q}$, we use PSNR, SSIM, LPIPS (Zhang et al., 2018a), FID (Heusel et al., 2017) metrics. Similar to (Wei et al., 2021), we use the AlexNet-based (Krizhevsky et al., 2012) LPIPS. FID and LPIPS are practically the *most important* since they better correlate with the human perception of the image quality. We include PSNR, SSIM as popular evaluation metrics, but they are known to *badly measure perceptual quality* (Zhang et al., 2018a; Nilsson & Akenine-Möller, 2020). Due to this, higher PSNR, SSIM values do not necessarily mean better performance. We calculate metrics using scikit-image for SSIM and open source implementations for PSNR[2], LPIPS[3] and FID[4]. In this section, we additionally use the $\mathcal{L}^2$-UVP (Korotin et al., 2021, §4.2) metric.

On the benchmark, we compare OTS (12) and IPM GAN (5). We use MSE as the content loss $c(x,y)$. In IPM GAN, we use the Wasserstein-1 ($\mathbb{W}_1$) loss with the gradient penalty $\lambda_{\text{GP}} = 10$ (Gulrajani et al., 2017) as $\mathcal{D}$. We do 10 discriminator updates per 1 generator update and train the model for 15K generator updates. For fair comparison, the rest hyperparameters match those of our algorithm.

---

[1]github.com/milesial/Pytorch-UNet
[2]github.com/photosynthesis-team/piq
[3]github.com/richzhang/PerceptualSimilarity
[4]github.com/mseitzer/pytorch-fid

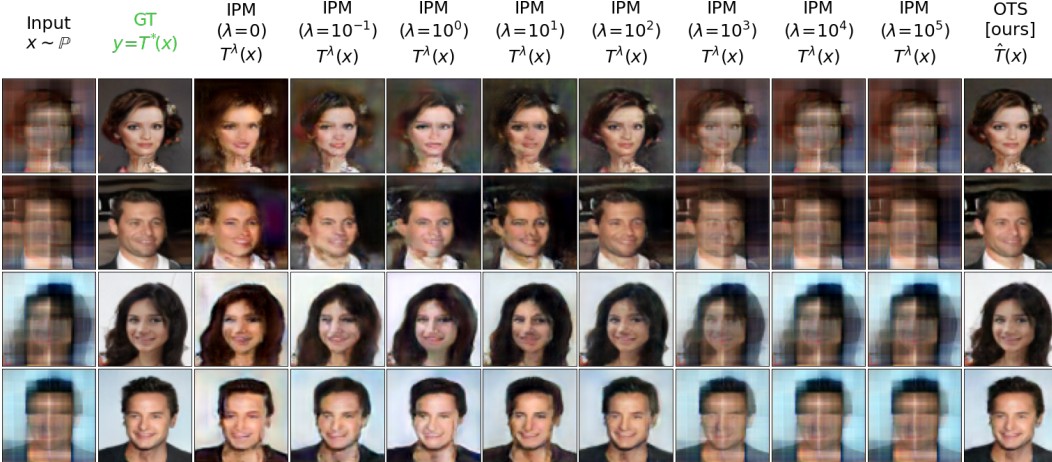

Figure 5: Comparison of OTS (ours), regularized IPM GAN on the Wasserstein-2 benchmark. The 1st line shows blurry faces $x \sim \mathbb{P}$, the 2nd line, clean faces $y = T^*(x)$, where $T^*$ is the OT map from $\mathbb{P}$ to $\mathbb{Q}$. Next lines show maps from $\mathbb{P}$ to $\mathbb{Q}$ fitted by the methods.

| Metrics/ Method | Regularized IPM GAN (WGAN-GP, $\lambda_{GP} = 10$) | | | | | | | | OTS (ours) |
|---|---|---|---|---|---|---|---|---|---|
| | $\lambda = 0$ | $\lambda = 10^{-1}$ | $\lambda = 10^0$ | $\lambda = 10^1$ | $\lambda = 10^2$ | $\lambda = 10^3$ | $\lambda = 10^4$ | $\lambda = 10^5$ | |
| $\mathcal{L}^2$-UVP $\downarrow$ | 25.2% | 16.7% | 17.7% | 12.0% | **4.0**% | 14.0% | 28.5% | 30.5% | **1.4**% |
| FID $\downarrow$ | 57.24 | 46.23 | 40.04 | 42.89 | **24.25** | 187.95 | 332.7 | 334.7 | **15.65** |
| PSNR $\uparrow$ | 17.90 | 19.76 | 19.34 | 20.81 | **25.58** | 19.91 | 16.90 | 16.52 | **30.02** |
| SSIM $\uparrow$ | 0.565 | 0.655 | 0.656 | 0.689 | **0.859** | 0.702 | 0.520 | 0.498 | **0.933** |
| LPIPS $\downarrow$ | 0.135 | 0.093 | 0.099 | 0.081 | **0.031** | 0.172 | 0.429 | 0.446 | **0.013** |

Table 2: Quantitative evaluation of restoration maps fitted by the regularized IPM GAN, OTS (ours) using the Wasserstein-2 images benchmark (Korotin et al., 2021).

We train the regularized WGAN-GP with various coefficients of content loss $\lambda \in \{0, 10^{-1}, \ldots, 10^5\}$ and show the learned maps $T^\lambda$ and the map $\hat{T}$ obtained by OTS in Figure 5.

**Results.** The performance of the regularized IPM GAN *significantly* depends on the choice of the content loss value $\lambda$. For high values $\lambda \geq 10^3$, the learned map is close to the identity as expected. For small values $\lambda \leq 10^1$, the regularization has little effect, and WGAN-GP solely struggles to fit a good restoration map. Even for the best performing $\lambda = 10^2$ all metrics are notably worse than for OTS. Importantly, *OTS decreases the burden of parameter searching* as there is no parameter $\lambda$.

### 7.2 LARGE-SCALE EVALUATION

For evaluating our method at a large-scale, we employ the dataset by (Lugmayr et al., 2019b) of AIM 2019 Real-World Super-Resolution Challenge (Track 2). The train part contains 800 HR images with up to 2040 pixels width or height and 2650 unpaired LR images of the same shape. They are constructed using artificial, but realistic, image degradations. We quantitatively evaluate our method on the validation part of AIM dataset that contains 100 pairs of LR-HR images.

**Baselines.** We compare OTS on AIM dataset with the bicubic upsample, FSSR (Fritsche et al., 2019) and DASR (Wei et al., 2021) methods. FSSR method is the winner of AIM 2019 Challenge; DASR is a current state-of-the-art method for unpaired image SR. Both methods utilize the idea of frequency separation and solve the problem in two steps. First, they train a network to generate LR images. Next, they train a super-resolution network using generated pseudo-pairs. Differently to FSSR, DASR also employs real-world LR images for training SR network taking into consideration the domain gap between generated and real-world LR images. Both methods utilize several losses, e.g., adversarial and perceptual, either on the entire image or on its high/low frequency components. For testing FSSR and DASR, we use their official code and pretrained models.

**Implementation details.** We train the networks using $128 \times 128$ HR, $32 \times 32$ LR random *patches* of images augmented via random flips, rotations. We conduct separate experiments using EDSR as the transport map and either MSE or perceptual cost, and denote them as OTS (MSE), OTS (VGG) respectively.

**Metrics.** We calculate PSNR, SSIM, LPIPS, FID. FID is computed on $32 \times 32$ patches of LR test images upsampled by the method in view w.r.t. random patches of test HR. We use 50k patches to compute FID. The other metrics are computed on the *entire* upsampled LR test and HR test images.

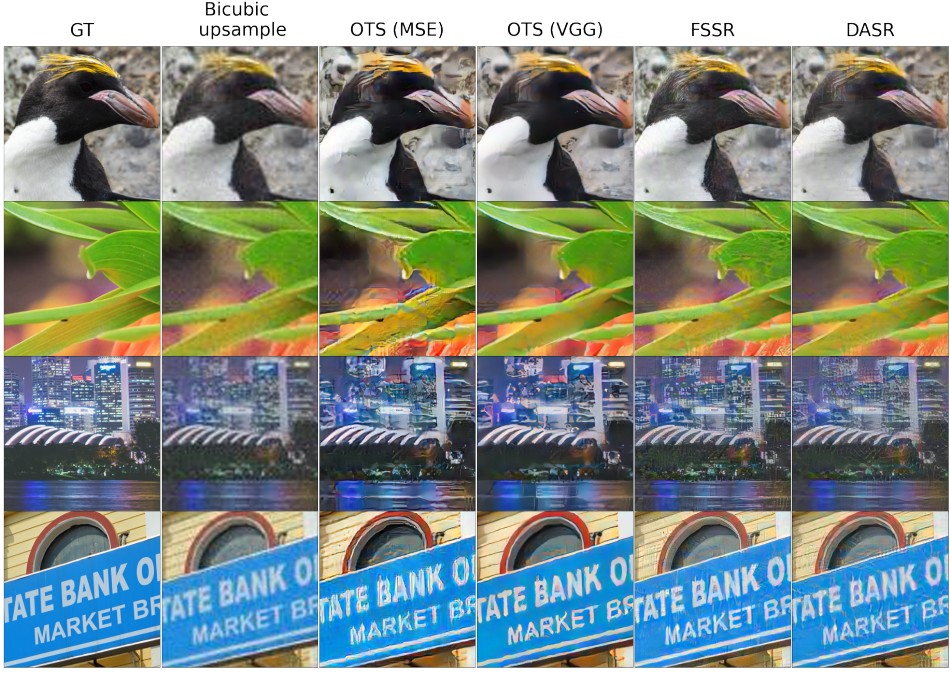

Figure 6: Qualitative results of OTS (ours), bicubic upsample, FSSR and DASR on AIM 2019 dataset (350×350 crops).

**Experimental results** are given in Table 3, Figure 6. The results show that the usage perceptual cost function in OTS boosts performance. According to FID, OTS with perceptual cost function beats DASR. On the other hand, it outperforms FSSR in PSNR, SSIM and, importantly, LPIPS. Note that bicubic upsample outperforms all the methods, according only to PSNR and SSIM, which have issues stated in §7.1. According to visual analysis, OTS with the perceptual cost better deals with noise artifacts. Additional results are given in Appendix E. We also demonstrate the bias issue of FSSR and DASR in Appendix B.

| Method | FID ↓ | PSNR ↑ | SSIM ↑ | LPIPS ↓ |
|---|---|---|---|---|
| Bicubic upsample | 178.59 | 22.39 | 0.613 | 0.688 |
| OTS (MSE) | 139.17 | 19.73 | 0.533 | 0.456 |
| OTS (VGG) | 89.04 | 20.96 | 0.605 | 0.380 |
| FSSR | 53.92 | 20.83 | 0.514 | 0.390 |
| DASR | 124.09 | 21.79 | 0.577 | 0.346 |

Table 3: Comparison of OTS (ours) with FSSR, DASR on AIM19 dataset. The 1st, 2nd, 3rd best results are highlighted in green, blue and underlined, respectively.

## 8 DISCUSSION

**Significance.** Our analysis connects content losses in GANs with OT and reveals the bias issue. Content losses are used in a wide range of tasks besides SR, e.g., in the style transfer and domain adaptation tasks. Our results demonstrate that GAN-based methods in all these tasks may *a priori lead to biased solutions*. In certain cases it is undesirable, e.g., in medical applications (Bissoto et al., 2021). Failing to learn true data statistics (and learning biased ones instead), e.g., in the super-resolution of MRI images, might lead to a wrong diagnosis made by a doctor due to SR algorithm drawing **inexistent details** on the scan. Thus, we think it is essential to emphasize and alleviate the bias issue, and provide a way to circumvent this difficulty.

**Potential Impact.** We expect our OT approach to improve the existing applications of image super-resolution. Importantly, it has less hyperparameters, uses smaller number of neural networks than many existing methods (see Table 5 in Appendix C for comparison), and is end-to-end — this should simplify its usage in practice. Besides, our method is generic and presumably can be applied to other unpaired learning tasks as well. Studying such applications is a promising avenue for the future work.

**Limitations.** Our method fits a one-to-one optimal mapping (transport map) for super-resolution which, in general, might not exist. Besides, not all optimal solutions of our optimization objective are guaranteed to be OT maps. Moreover, our method requires solving the saddle point problem. Thus, it might encounter training issues similar to those of the GAN-based approaches. These limitations suggest the need for further theoretical analysis and improvement of our method for optimal transport.

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

## A First Variations of GAN Discrepancies Vanish at the Optimum

We demonstrate that the first variation of $\mathbb{Q}' \mapsto \mathcal{D}(\mathbb{Q}', \mathbb{Q})$ is equal to zero at $\mathbb{Q}' = \mathbb{Q}$ for common GAN discrepancies $\mathcal{D}$. This suggests that the corresponding assumption of our Theorem 1 is relevant.

To begin with, for a functional $\mathcal{G} : \mathcal{P}(\mathcal{Y}) \to \mathbb{R} \cup \{\infty\}$, we recall the definition of its **first variation**. A measurable function $\delta\mathcal{G}[\mathbb{Q}] : \mathcal{Y} \to \mathbb{R} \cup \{\infty\}$ is called **the first variation** of $\mathcal{G}$ at a point $\mathbb{Q} \in \mathcal{P}(\mathcal{Y})$, if, for every measure $\Delta\mathbb{Q}$ on $\mathcal{Y}$ with zero total mass ($\int_{\mathcal{Y}} 1 \, d\Delta\mathbb{Q}(y) = 0$),

$$\mathcal{G}(\mathbb{Q} + \epsilon\Delta\mathbb{Q}) = \mathcal{G}(\mathbb{Q}) + \epsilon \int_{\mathcal{Y}} \delta\mathcal{G}[\mathbb{Q}](y) \, d\Delta\mathbb{Q}(y) + o(\epsilon) \tag{16}$$

for all $\epsilon \geq 0$ such that $\mathbb{Q} + \epsilon\Delta\mathbb{Q}$ is a probability distribution. Here for the sake of simplicity we suppressed several minor technical aspects, see (Santambrogio, 2015, Definition 7.12) for details. Note that the first variation is defined **up to an additive constant**.

Now we recall the definitions of three most popular GAN discrepancies and demonstrate that their first variation is zero at an optimal point. We consider $f$-divergences (Nowozin et al., 2016), Wasserstein distances (Arjovsky et al., 2017), and Maximum Mean Discrepancies (Li et al., 2017).

**Case 1** ($f$-divergence). Let $f : \mathbb{R}_+ \to \mathbb{R}$ be a convex and differentiable function satisfying $f(1) = 0$. The $f$-divergence between $\mathbb{Q}', \mathbb{Q} \in \mathcal{P}(\mathcal{Y})$ is defined by

$$\mathcal{D}_f(\mathbb{Q}', \mathbb{Q}) \overset{def}{=} \int_{\mathcal{Y}} f\left(\frac{d\mathbb{Q}'(y)}{d\mathbb{Q}(y)}\right) d\mathbb{Q}(y). \tag{17}$$

The divergence takes finite value only if $\mathbb{Q}' \ll \mathbb{Q}$, i.e., $\mathbb{Q}'$ is absolutely continuous w.r.t. $\mathbb{Q}$. Vanilla GAN loss (Goodfellow et al., 2014) is a case of $f$-divergence (Nowozin et al., 2016, Table 1).

We define $\mathcal{G}(\mathbb{Q}') \overset{def}{=} \mathcal{D}_f(\mathbb{Q}', \mathbb{Q})$. For $\mathbb{Q}' = \mathbb{Q}$ and some $\Delta\mathbb{Q}$ such that $\mathbb{Q} + \epsilon\Delta\mathbb{Q} \in \mathcal{P}(\mathcal{Y})$ we derive

$$\mathcal{G}(\mathbb{Q} + \epsilon\Delta\mathbb{Q}) = \int_{\mathcal{Y}} f\left(\frac{d\mathbb{Q}(y)}{d\mathbb{Q}(y)} + \epsilon\frac{d\Delta\mathbb{Q}(y)}{d\mathbb{Q}(y)}\right) d\mathbb{Q}(y) = \int_{\mathcal{Y}} f\left(1 + \epsilon\frac{d\Delta\mathbb{Q}(y)}{d\mathbb{Q}(y)}\right) d\mathbb{Q}(y) \tag{18}$$

$$= \int_{\mathcal{Y}} f(1) d\mathbb{Q}(y) + \int_{\mathcal{Y}} f'(1)\frac{d\Delta\mathbb{Q}(y)}{d\mathbb{Q}(y)} d\mathbb{Q}(y) + o(\epsilon) = \mathcal{G}(\mathbb{Q}) + \int_{\mathcal{Y}} f'(1) d\Delta\mathbb{Q}(y) + o(\epsilon), \tag{19}$$

where in transition from (18) to (19), we consider the Taylor series w.r.t. $\epsilon$ at $\epsilon = 0$. We see that $\delta\mathcal{G}[\mathbb{Q}](y) \equiv f'(1)$ is constant, i.e., the first variation of $\mathbb{Q}' \mapsto \mathcal{D}_f(\mathbb{Q}', \mathbb{Q})$ vanishes at $\mathbb{Q}' = \mathbb{Q}$.

**Case 2** (Wasserstein distance). If in OT formulation (3) the cost function $c(x, y)$ equals $\|x - y\|^p$ with $p \geq 1$, then $\left[\text{Cost}(\mathbb{P}, \mathbb{Q})\right]^{1/p}$ is called the *Wasserstein distance* ($\mathbb{W}_p$). Generative models which use $\mathbb{W}_p^p$ as the discrepancy are typically called the Wasserstein GANs (WGANs). The most popular case is $p = 1$ (Arjovsky et al., 2017; Gulrajani et al., 2017), but more general cases appear in related work as well, see (Liu et al., 2019; Mallasto et al., 2019).

The first variation of $\mathcal{G}(\mathbb{Q}') \overset{def}{=} \mathbb{W}_p^p(\mathbb{Q}', \mathbb{Q})$ at a point $\mathbb{Q}'$ is given by $\mathcal{G}[\mathbb{Q}'](y) = (f^*)^c(y)$, where $f^*$ is the optimal dual potential (provided it is unique up to a constant) in (4) for a pair $(\mathbb{Q}', \mathbb{Q})$, see (Santambrogio, 2015, §7.2). Our particular interest is to compute the optimal potential $(f^*)^c$ at $\mathbb{Q}' = \mathbb{Q}$. We recall (4) and use $\mathbb{W}_p^p(\mathbb{Q}, \mathbb{Q}) = 0$ to derive

$$\mathbb{W}_p^p(\mathbb{Q}, \mathbb{Q}) = 0 = \sup_f \left[ \int_{\mathcal{X}} f^c(y') d\mathbb{Q}'(y') + \int_{\mathcal{Y}} f(y) d\mathbb{Q}(y) \right].$$

One may see that $f^* \equiv 0$ attains the supremum (its $c$-transform $(f^*)^c$ is also zero). Thus, if $(f^*)^c \equiv 0$ is a unique potential (up to a constant), the first variation of $\mathbb{Q}' \mapsto \mathbb{W}_p^p(\mathbb{Q}', \mathbb{Q})$ at $\mathbb{Q}' = \mathbb{Q}$ vanishes.

**Case 3** (Maximum Mean Discrepancy). Let $k : \mathcal{Y} \times \mathcal{Y} \to \mathbb{R}$ be a positive definite symmetric kernel. The (square) of the Maximum Mean Discrepancy between $\mathbb{Q}', \mathbb{Q}$ is given by

$$\text{MMD}_k^2(\mathbb{Q}', \mathbb{Q}) \overset{def}{=} \int_{\mathcal{Y} \times \mathcal{Y}} k(y_1, y_2) d\left[(\mathbb{Q} - \mathbb{Q}') \times (\mathbb{Q} - \mathbb{Q}')\right](y_1, y_2)$$

$$= \int_{\mathcal{Y} \times \mathcal{Y}} k(y_1, y_2) d(\mathbb{Q}' \times \mathbb{Q}')(y_1, y_2) - 2 \int_{\mathcal{Y} \times \mathcal{Y}} k(y_1, y_2) d(\mathbb{Q}' \times \mathbb{Q})(y_1, y_2) + \text{Const}(\mathbb{Q}) \tag{20}$$

see (Sejdinovic et al., 2013, Equation 3.3). The first variation of the quadratic in $\mathbb{Q}'$ term is given by $y \mapsto 2 \cdot \int_{\mathcal{Y}} k(y, y_2) d\mathbb{Q}'(y_2)$, see (Santambrogio, 2015, §7.2). The second term is linear in $\mathbb{Q}'$ and its first variation is simply $y \mapsto (-2) \cdot \int_{\mathcal{Y}} k(y, y_2) d\mathbb{Q}(y_2)$. When $\mathbb{Q}' = \mathbb{Q}$, the sum of these terms is zero. That is, the first variation of the functional $\mathbb{Q}' \mapsto \text{MMD}_k^2(\mathbb{Q}', \mathbb{Q})$ vanishes at $\mathbb{Q}' = \mathbb{Q}$.

## B ASSESSING THE BIAS OF METHODS ON AIM19 DATASET

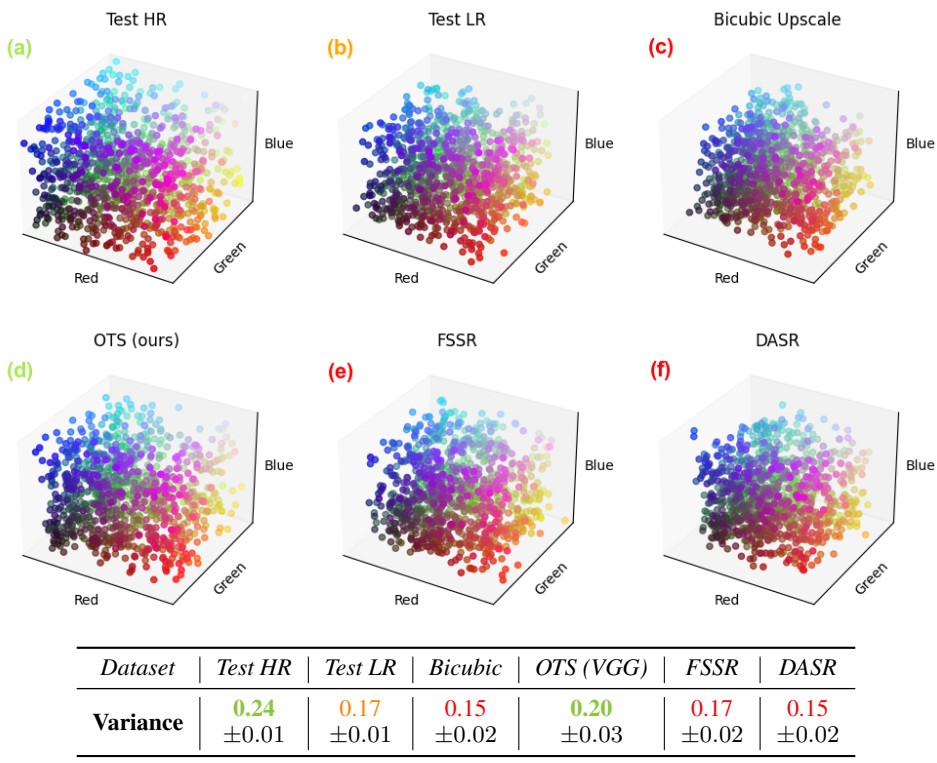

| Dataset | Test HR | Test LR | Bicubic | OTS (VGG) | FSSR | DASR |
|---------|---------|---------|---------|-----------|------|------|
| **Variance** | **0.24** $\pm 0.01$ | 0.17 $\pm 0.01$ | 0.15 $\pm 0.02$ | **0.20** $\pm 0.03$ | 0.17 $\pm 0.02$ | 0.15 $\pm 0.02$ |

Figure 7: Color palettes and their variance for Test HR, LR datasets and solutions of Bicubic Upscale, OTS, FSSR, DASR methods on AIM19.

We additionally demonstrate the bias issue by comparing color palettes of HR images and super-resolution results of different methods, see Figure 7. We construct palettes by choosing random image pixels from dataset images and representing them as an RGB point cloud in $[0, 1]^3 \subset \mathbb{R}^3$. Figure 7 shows that OTS **(d)** captures *large contrast* of HR **(a)** images (variance of its palette), while FSSR **(e)**, DASR **(f)**, Bicubic Upscale **(c)** palettes are *less contrastive* and closer to LR **(b)**. We construct palettes 100 times to evaluate their average contrast (variance). The metric *quantitatively* confirms that our OTS method better captures the contrast of HR dataset, while GAN-based methods (FSSR and DASR) are notably *biased* towards LR dataset statistics (low contrast).

## C TRAINING DETAILS

**Perceptual cost.** In 7.2 we test following *perceptual cost* as $b$:

$$b(x^{\text{up}}, y) = \text{MSE}(x^{\text{up}}, y) + 1/3 \cdot \text{MAE}(x^{\text{up}}, y) + 1/50 \cdot \sum_{k \in \{3,8,15,22\}} \text{MSE}(f_k(x^{\text{up}}), f_k(y)),$$

where $f_k$ denotes the features of the $k$th layer of a pre-trained VGG-16 network (Simonyan & Zisserman, 2014), MAE is the mean absolute error $\text{MAE}(x, y) = \frac{\|x-y\|_1}{\dim(\mathcal{Y})}$.

**Dynamic transport cost**. In the preliminary experiments, we used bicubic upsampling as the "Up" operation. Later, we found that the method works better if we gradually change the upsampling. We start from the bicubic upsampling. Every $k_c$ iterations of $f_\omega$ (see Table 4), we change the cost to $c(x, y) = b(T'_\theta(x), y)$, where $T'_\theta$ is a fixed frozen copy of the currently learned SR map $T_\theta$.

**Hyperparameters.** For EDSR, we set the number of residual blocks to 64, the number of features to 128, and the residual scaling to 1. For UNet, we set the base factor to 64. The training details are given in Table 4. We provide a comparison of the hyperparameters of FSSR, DASR and OTS (ours) in Table 5. In contrast to FSSR and DASR, our method does not contain a degradation part. This helps to notably reduce the amount of tunable hyperparameters.

**Optimizer.** We employ Adam (Kingma & Ba, 2014).

**Computational complexity**. Training OTS with EDSR as the transport map and the perceptual transport cost on AIM 2019 dataset takes $\approx 4$ days on a single Tesla V100 GPU.

| Experiment | dim($\mathcal{X}$) | dim($\mathcal{Y}$) | $f$ | $T$ | $k_T$ | $lr_f$ | $lr_T$ | Initial cost | Total iters ($f$) | Cost update every | Batch size |
|---|---|---|---|---|---|---|---|---|---|---|---|
| Benchmark (§7.1) | $3 \times 64 \times 64$ | $3 \times 64 \times 64$ | | UNet | 10 | | | MSE | 10K | — | 64 |
| Celeba (§D) | $3 \times 16 \times 16$ | $3 \times 64 \times 64$ | ResNet | Bilinear + UNet | 15 | $10^{-4}$ | $10^{-4}$ | Bicubic + MSE | 100K | 25K | 64 |
| | | | | EDSR | 15 | | | | 100K | 25K | 64 |
| AIM-19 (§7.2) | $3 \times 32 \times 32$ (patches) | $3 \times 128 \times 128$ (patches) | | EDSR | 15 | | | | 50K | 25K | 8 |
| | | | | EDSR | 10 | | | Bicubic + VGG | 50K | 20K | 8 |

Table 4: Hyperparameters that we use in the experiments with our Algorithm 1.

| Method | Degradation part | Super-resolution part | Total |
|---|---|---|---|
| **FSSR** | 2 neural networks;
2 optimizers;
2 schedulers;
1 adversarial loss;
1 content loss ($\ell_1$+perceptual) | 2 neural networks;
2 optimizers;
2 schedulers;
1 adversarial loss;
1 content loss ($\ell_1$+perceptual) | 4 neural networks;
4 optimizers;
4 schedulers;
2 adversarial losses;
2 content losses ($\ell_1$+perceptual) |
| **DASR** | 2 neural networks;
2 optimizers;
2 schedulers;
1 adversarial loss;
1 content loss ($\ell_1$+perceptual) | 2 neural networks;
2 optimizers;
2 schedulers;
1 adversarial loss;
1 content loss ($\ell_1$+perceptual) | 4 neural networks;
4 optimizers;
4 schedulers;
2 adversarial losses;
2 content losses ($\ell_1$+perceptual) |
| **OTS (ours)** | — | 2 neural networks;
2 optimizers;
1 cost ($\ell_2$+$\ell_1$+perceptual) | 2 neural networks;
2 optimizers;
1 cost ($\ell_2$+$\ell_1$+perceptual) |

Table 5: Comparison of hyperparameters used in FSSR, DASR and OTS (ours) methods.

# D  IMAGE SUPER-RESOLUTION OF FACES

We conduct an experiment using CelebA (Liu et al., 2015) faces to test the applicability of OT for unpaired SR. We test our Algorithm 1 with MSE as the cost and UNet or EDSR as the transport map.

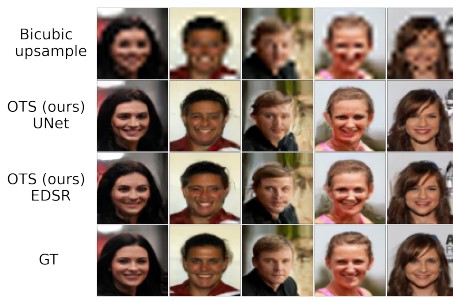

Figure 8: Qualitative results of OTS (ours) on CelebA.

| Method | FID ↓ | PSNR ↑ | SSIM ↑ | LPIPS ↓ |
|---|---|---|---|---|
| Bicubic upsample | 130.72 | 22.73 | 0.756 | 0.303 |
| OTS (ours) UNet | 12.32 | 22.10 | 0.740 | 0.058 |
| OTS (ours) EDSR | 15.87 | 22.33 | 0.747 | 0.054 |

Table 6: Comparison of OTS (ours) with the bicubic upsampling on CelebA dataset. The 1st and 2nd best results are highlighted in green and blue, respectively.

**Pre-processing and train-test split.** We resize images to $64 \times 64$ px. We adopt the *unpaired* train-test split from (Rout et al., 2022, §5.2). We split the original HR dataset in 3 parts A, B, C containing 90K, 90K, 22K samples, respectively. We apply the bicubic downsample to each image and obtain the LR dataset ($16 \times 16$ faces). For training, we use LR part A, HR part B. For testing, we use parts C.

**Metrics.** We compute PSNR, SSIM, LPIPS and FID metrics on the *test* part, see Table 6.

# E   ADDITIONAL RESULTS ON AIM19

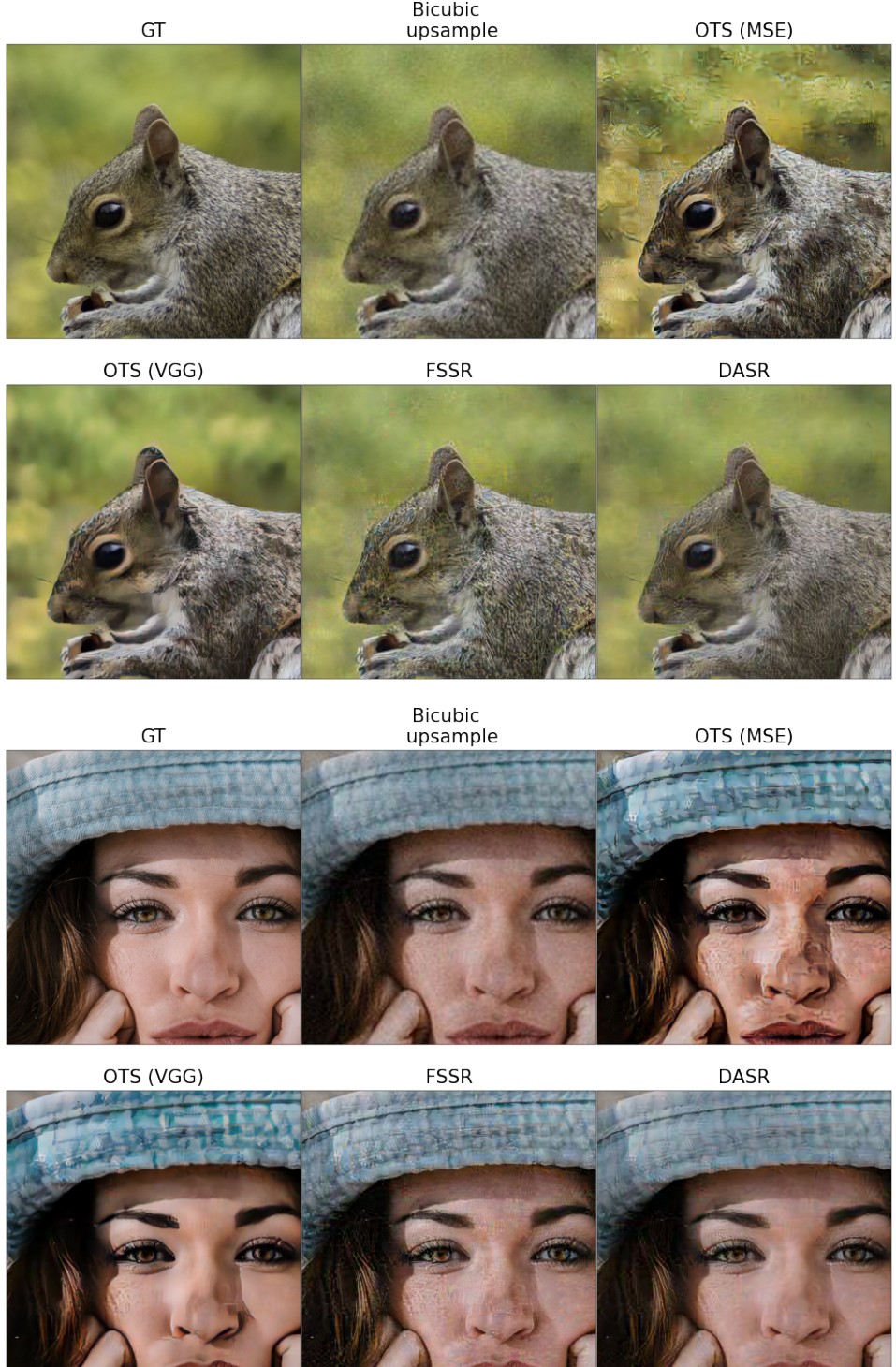

Figure 9: Additional qualitative results of OTS (ours), bicubic upsample, FSSR and DASR on AIM 2019 (800×800 crops).

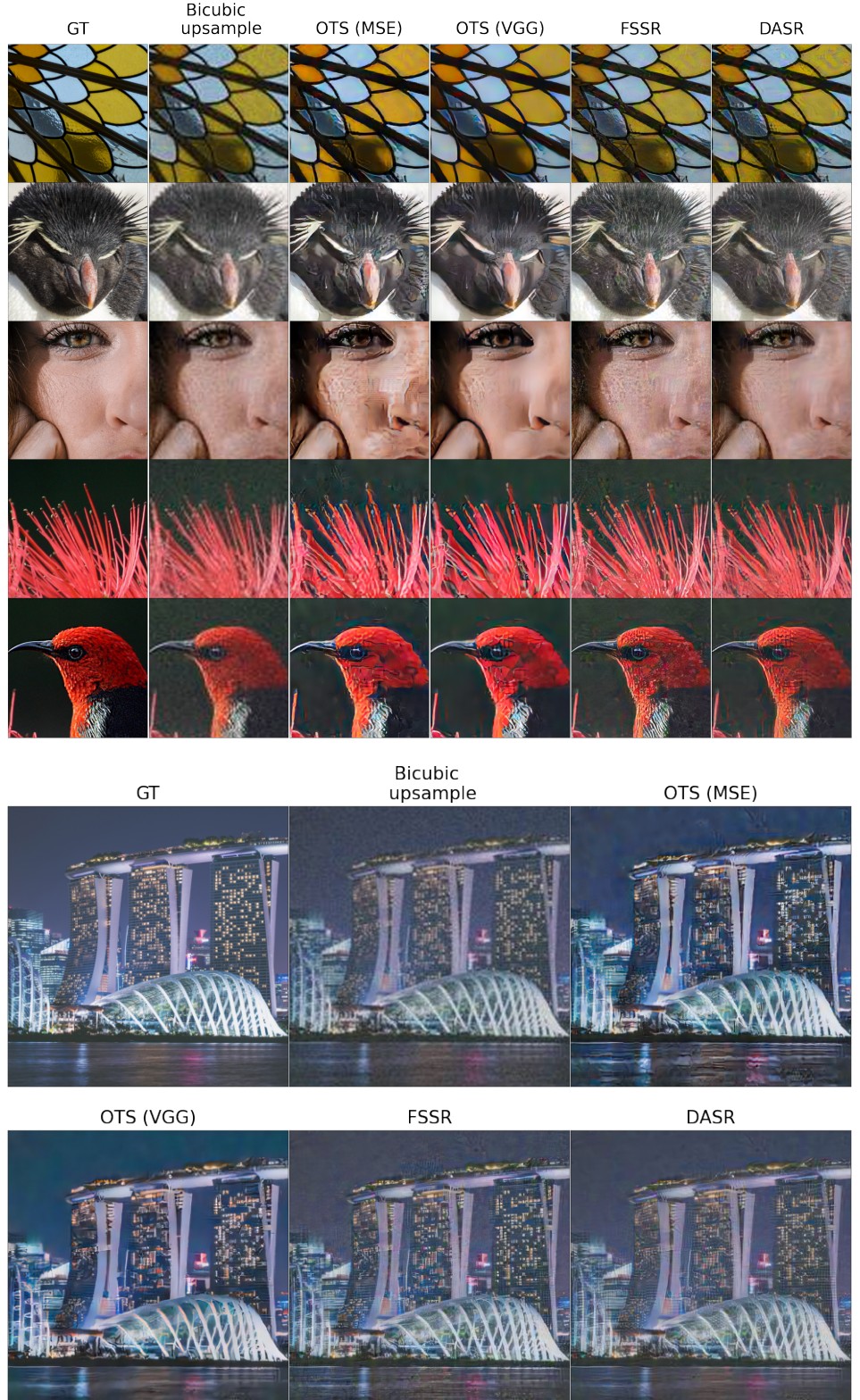

Figure 10: Additional qualitative results of OTS (ours), bicubic upsample, FSSR and DASR on AIM 2019. The sizes of crops on the 1st and 2nd images are 350×350 and 800×800, respectively.

## F    CONNECTION BETWEEN GAN OBJECTIVES AND EQUATION (5)

Typical objectives of GAN-based approaches consist of multiple losses − usually one adversarial and several content losses. To make the exposition simple, in our paper, we represented all the content losses as a single loss $c(\cdot, \cdot)$. Below we provide several examples showing how the objectives of popular GAN-based approaches to unpaired image SR could be viewed as (5). For all of these methods, our Lemma 1 applies without any changes. We include in brackets the number of papers citations according to Google Scholar to show that chosen methods are widely used.

**FaceSR** (2018, 284 citations) The paper of (Bulat et al., 2018) presents one of the first GAN-based approaches to unpaired image SR problem. The method is composed of two steps. First, it learns a degradation between unpaired HR and LR images. Then it employs a second GAN to learn a supervised mapping between paired generated LR and corresponding HR images. The objective of the unpaired step (see their Eq. (1)) is as follows:

$$l = \underbrace{\alpha l_{\text{pixel}}}_{\text{content loss}} + \underbrace{\beta l_{\text{GAN}}}_{\text{adversarial loss}}.$$

Here $l_{\text{pixel}}$ is the MSE loss between the generated LR image and downsampled HR. Thus, the objective of this method exactly follows Equation (5).

**CinCGAN** (2018, 400 citations) The method of (Yuan et al., 2018) is an other pioneering GAN-based approach to unpaired image SR problem, which establishes a different to FaceSR group of two-step methods. First, it uses one CycleGAN to learn a mapping between given noisy LR images and downsampled HR ("clean LR") images. Then, a second CycleGAN fine-tunes a mapping between real LR and HR images. The objective for the first GAN (see their Eq. (5)) is as follows:

$$\mathcal{L}_{\text{total}}^{LR} = \underbrace{\mathcal{L}_{\text{GAN}}^{\text{LR}}}_{\text{adversarial loss}} + \underbrace{w_1 \mathcal{L}_{\text{cyc}}^{\text{LR}} + w_2 \mathcal{L}_{\text{idt}}^{\text{LR}} + w_3 \mathcal{L}_{\text{TV}}^{\text{LR}}}_{\text{content loss}}.$$

Here $\mathcal{L}_{\text{cyc}}^{LR}$ is the cycle-consistency loss[5], $\mathcal{L}_{\text{idt}}^{\text{LR}} - l_1$ identity loss, and $\mathcal{L}_{\text{TV}}^{\text{LR}} -$ total variation loss.

**FSSR** (Winner of the AIM Challenge on Real-World SR (Lugmayr et al., 2019b), 2019, 127 citations) FSSR (Fritsche et al., 2019) method employs a similar to FaceSR strategy. It firstly learns a mapping between downsampled HR images and given unpaired LR images, and then uses the generated pairs to learn a supervised SR model. The objective of the unpaired step (see their Eq. (6)) is defined by:

$$\mathcal{L}_d = \underbrace{0.005 \mathcal{L}_{\text{tex, d}}}_{\text{adversarial loss}} + \underbrace{\mathcal{L}_{\text{col, d}} + 0.01 \mathcal{L}_{\text{per, d}}}_{\text{content loss}},$$

where the texture (adversarial) loss $\mathcal{L}_{\text{tex, d}}$ and the color ($l_1$ identity) loss $\mathcal{L}_{\text{col, d}}$ are applied to low frequencies of the images, while the perceptual loss $\mathcal{L}_{\text{per, d}} -$ to the features of the full images.

**DASR** (2021, 51 citations) DASR (Wei et al., 2021) structure is also based on the similar to FSSR principles and its two-step structure. In contrast to FSSR, a SR network is trained in a partially supervised manner using not only generated, but also real LR images. The objective of the fully unpaired degradation learning step (see their Eq. (4)) is as follows:

$$\mathcal{L}_{\text{DSN}} = \underbrace{\alpha \mathcal{L}_{\text{con}} + \beta \mathcal{L}_{\text{per}}}_{\text{content loss}} + \underbrace{\gamma \mathcal{L}_{\text{adv}}^G}_{\text{adversarial loss}}.$$

Here the adversarial loss $\mathcal{L}_{\text{adv}}^G$ is defined on high frequencies of the image, while the content $\mathcal{L}_{\text{con}}$ ($l_1$ identity) and the perceptual $\mathcal{L}_{\text{per}}$ losses are defined on full images and their features respectively.

**ESRGAN-FS** (2020, 13 citations) ESRGAN-FS is an other two-step approach based on the principle of learning the degradation, see (Zhou et al., 2020). The objective of its unpaired degradation learning step (see their Eq. (4)) is as follows:

$$\mathcal{L}_{\text{total}} = \underbrace{\lambda_{t1} \cdot \mathcal{L}_{\text{low}} + \lambda_{t2} \cdot \mathcal{L}_{\text{per}}}_{\text{content loss}} + \underbrace{\lambda_{t3} \cdot \mathcal{L}_{\text{high}}}_{\text{adversarial loss}}.$$

Here $\mathcal{L}_{\text{low}}$ ($l_1$ identity) loss is applied to low frequencies of the images, the perceptual loss $\mathcal{L}_{\text{per}} -$ to the features of the full images, while $\mathcal{L}_{\text{high}}$ (adversarial loss) − high frequencies of the images.

---

[5]$\mathcal{L}_{\text{cyc}}^{\text{LR}}$ is defined as the MSE loss between given LR image $x$ and $G_2(G_1(x))$, where $G_1$ learns to map real LR images to "clean" ones and $G_2$ learns an opposite mapping. For a fixed $G_2$ this loss can be considered as a part of the content loss.

