# OpenReview forum: "An Optimal Transport Perspective on Unpaired Image Super-Resolution"
_ICLR.cc/2023/Conference — Submitted to ICLR 2023_

### Official Review · Reviewer_nWwQ · 2022-10-23

**Confidence:** 3
**Correctness:** 4
**Technical Novelty And Significance:** 3
**Empirical Novelty And Significance:** 3
**Recommendation:** 8

**Clarity, Quality, Novelty And Reproducibility:**

The work is very well written, the problem statement, the mathematical formulation, key lemmas and main proofs are very clean the elegant, the deduction is easy to follow, the experimental results are well analyzed and convincing. The observations are novel, it discovers the intrinsic connection between SR problem and OT theory. The authors provide source codes, it is easy to reproduce.

**Details Of Ethics Concerns:**

This work focuses on fundamental research, it focuses on theoretical aspect. Furthermore, it points out that conventional method may produce biased medical images, which may lead to wrong diagnosis. So this work improves the conventional method and overcome the problem.

**Strength And Weaknesses:**

The strength of the work are:  it has unique observations on conventional GANs for unpaired SR problem, and proves the solutions of GANs are OT maps, but biased, then provides an unbiased solution; The mathematical model and formulations are elegant and powerful, the algorithm is effective; the experimental results show the proposed method achieves SOTA in SR applications; it is a good example of combing rigorous mathematical model with real world applications.

The weakness is that the minmax optimization proposed by the work can find the saddle points, which may not correspond to optimal solutions, this step needs further exploration.

**Summary Of The Paper:**

This work theoretically investigate the unpaired techniques based on Generative Adversarial Networks (GANs) which arise in  super-resolution tasks, and find two observations: first the learned SR map is always an optimal transport map; second, the learned map is biased. The observations are supported by theoretical proofs and empirical experiments. The work proposes an algorithm for unpaired SR which learns an unbaised OT map for the perceptual transport cost. This method reduces the need for complex hyperparamter selection and additional regularizations. The proposed model provides a nearly state-of-the-art performance on unpaired AIM19 datasest.

The contributions are:
1. the work investigate the GAN optimization objectives regularized with content loss, and prove the solutions are always OT maps, but biased;
2. the work provide an algorithm to fit an unbiased OT map for perceptual transport cost and applied it for the SR problem, establishes connections between the proposed model and regularized GANs using IPMs as a loss.


**Summary Of The Review:**

This work theoretically investigate the unpaired techniques based on Generative Adversarial Networks (GANs) which arise in  super-resolution tasks, and find two observations: first the learned SR map is always an optimal transport map; second, the learned map is biased. The observations are supported by theoretical proofs and empirical experiments. The work proposes an algorithm for unpaired SR which learns an unbaised OT map for the perceptual transport cost. This method reduces the need for complex hyperparamter selection and additional regularizations. The proposed model provides a nearly state-of-the-art performance on unpaired AIM19 datasest.

These observations are novel, it discovers the intrinsic connection between SR problem and OT theory. The work is very well written, the problem statement, the mathematical formulation, key lemmas and main proofs are very clean the elegant, the deduction is easy to follow, the experimental results are well analyzed and convincing.  The authors provide source codes, it is easy to reproduce. In summary, the work is inspiring and promising, it has both theoretical and practical values.

---

> ### Author Response · Authors · 2022-11-16
> **Answer to Reviewer nWwQ**
>
> Dear reviewer, thanks for your positive review! We are inspired by the fact that your highly appreciate our contribution and find our approach inspiring and promising.
>
> We agree with your comment that the theoretical ambiguity in saddle points might require further exploration. Deriving sufficient conditions on distributions $\mathbb{P}$, $\mathbb{Q}$ and transport cost $c(x, y)$ to remove this ambiguity is a promising future research avenue.

---

### Official Review · Reviewer_mWTP · 2022-10-24

**Confidence:** 5
**Correctness:** 3
**Technical Novelty And Significance:** 3
**Empirical Novelty And Significance:** 2
**Recommendation:** 6

**Clarity, Quality, Novelty And Reproducibility:**

It is interesting to see that the authors expand the Dual form of OT to address unpaired SR task. For the technical side, the algirithom proposed in this paper extends the applications of existing SR practice.
And, the paper is well-organized and clearly written.
However, the final inequality of Equation10 requires more detailed proof. Intuitively, even for sufficiently small ϵ, the inequality is not obvious.

**Strength And Weaknesses:**

This paper has a sufficient theoretical analysis, but its weaknesses is that the empirical evaluation is not sufficient.
Only two methods FSSR (Fritsche et al., 2019) and DASR (Wei et al., 2021) have been compared on the dataset AIM 2019, and both methods use two GANs. There are many existing SR methods based on a single WGAN. What is the reason why this paper does not compare with them?
In addition, the following issues still need to be further explained.
1. As shown in equation 12, the proposed method fits a one-to-one optimal mapping (transport map) for super-resolution which, in general, might not exist.  To solve this problem, what special settings are required in real world super-resolution tasks?
2. The author declares that the proposed OTS is different from the regularized IPM GAN, Because in OST,  the OT map T is a solution to the inner optimization problem, while in IPM GAN the
generator T is a solution to the outer problem.  But from algorithm 1, there is no difference between finding inf first and finding sup first.
3. The author declares  ' in OTS the optimization over potential f is unconstrained, while in IPM GAN it must belong to F '.
     This is not fair.  On the one hand, both potential functions need to be realized by neural networks. On the other hand, for  regularized IPM GAN, distance measure is also optional, but the two algorithms referred to in this paper are  Wasserstein-1.

**Summary Of The Paper:**

This paper introduces  an algorithm for unpaired  super-resolution which learns an unbiased OT map for the perceptual transport cost.  Unlike the existing GAN-based alternatives, this proposed  algorithm has a simple optimization objective reducing the need for complex hyperparameter selection and an application of additional regularizations.

Contributions:
This paper provides an algorithm to fit an unbiased OT map for perceptual transport cost and apply it to the unpaired image SR problem. In addition, it investigate the GAN optimization objectives regularized with content losses.

**Summary Of The Review:**

This paper introduces  an algorithm for unpaired  super-resolution which learns an unbiased OT map for the perceptual transport cost.  This proposed algirithom extends the applications of existing SR practice.
The approach is technically relatively sound with some small flaw.  the paper is well-organized and clearly written.
More experimental evaluations are helpful to prove the effectiveness of the algorithm.

---

> ### Author Response · Authors · 2022-11-16
> **Answer to Reviewer mWTP**
>
> Thank you for your valuable feedback. Please find above (in our reply to all the Reviewers) the answers to your comments common with other reviews. Please find below our answers to your questions that do not overlap with those of other Reviewers.
>
> **(1) The empirical evaluation is not sufficient. Only two methods FSSR (Fritsche et al., 2019) and DASR (Wei et al., 2021) have been compared on the dataset AIM 2019,...**
>
> We choose to compare our method with FSSR and DASR approaches which demonstrate good performance, provide code and, importantly, pretrained models for the AIM19 dataset. Since the main focus of our paper is on theoretical exploration, we think that provided comparisons are sufficient.
>
> **(2) ... and both methods use two GANs. There are many existing SR methods based on a single WGAN. What is the reason why this paper does not compare with them?**
>
> We kindly ask the reviewer to state the method he/she wants us to compare with. To our knowledge, state-of-the-art GAN-based approaches to unpaired image SR are mostly based on two-GANs strategy.
>
> **(3) As shown in equation (12), the proposed method fits a one-to-one optimal mapping (transport map) for super-resolution which, in general, might not exist. To solve this problem, what special settings are required in real world super-resolution tasks?**
>
> Strictly speaking, the one-to-one SR maps exist if the degradation filter is invertible. However, these criterion is impossible to check in real-world scenarios when the LR and HR images are unpaired and degradation filter is unknown. However, as we have stated in Section 2 of our paper, most existing SR methods do not take this issue into account and learn a one-to-one mapping while still provide good performance.
>
> **(4) The author declares' in OTS the optimization over potential $f$ is unconstrained, while in IPM GAN it must belong to $\mathcal{F}'$. This is not fair.**
>
> **(4a) On the one hand, both potential functions need to be realized by neural networks.**
>
> We do not mention this aspect in our work since in ML usually any continuous variable (e.g., disriminator in GANs) is approximated by a neural network.
>
> **(4b) On the other hand, for regularized IPM GAN, distance measure is also optional, but the two algorithms referred to in this paper are Wasserstein-1.**
>
> We kindly ask you to clarify which algorithms you are talking about?
>
> **(5) The final inequality of Equation (10) requires more detailed proof. Intuitively, even for sufficiently small $\varepsilon$, the inequality is not obvious.**
>
> Could you please clarify which details of Equation (10) proof remain unclear?
>
> **Concluding remarks**. Please respond to our post to let us know if the clarifications above suitably address your concerns about our work. We are happy to address any remaining points during the discussion phase; if the responses above are sufficient, we kindly ask that you consider raising your score.

---

> > ### Author Response · Authors · 2022-12-02
> > **Looking forward to receiving your feedback**
> >
> > Dear Reviewer mWTP,
> >
> > We thank you for your review and greatly appreciate the time you spent reviewing our paper.
> >
> > The end of the rebuttal phase is approaching. We would be grateful if we could hear your feedback regarding our answers to the reviews. We are happy to address any remaining points during the remaining period.
> >
> > Thanks in advance,
> >
> > Paper authors

---

### Official Review · Reviewer_GsaW · 2022-10-24

**Confidence:** 4
**Clarity, Quality, Novelty And Reproducibility:** The paper is generally well written.
**Correctness:** 3
**Technical Novelty And Significance:** 3
**Empirical Novelty And Significance:** 3
**Recommendation:** 5

**Strength And Weaknesses:**

Strengths:
1) The relations between the standard GAN formulation/optimization and different versions of optimal-transport schemes with respect to the 'content' cost are very interesting and mostly new. Most importantly, in my opinion, is that they give a new way of thinking about and analyzing different approaches under this setup.
2) The introduction is written very well, in term of background, context and motivation. Also, the main derivations, while being very formal, are clearly and well explained.

Weaknesses:
1) The first observation regarding GAN (Lemma 1 - that it is a particular OT map) is pretty trivial, although interesting. I wouldn't agree that the main observation (Thm1 - that map/generator is biased towards the input) is surprising. In fact, the addition of the 'content' cost of closeness between the input and output obviously constrains the output in such a way, which is probably inevitable in the 'unpaired' setup.
2) There is a problem with the logic that follows Lemma 3 - It wrongly claims that "Lemma 3 states that one can solve a saddle point problem (12), obtain an optimal pair (f∗, T∗), and use T ∗ as an OT map from P, Q". It was shown that the true T* can be chosen as optimal along with the true f*, but this doesn't show that the map obtained along with f* is actually that one and not another. The following sentence "For general P, Q, the arg infT set for an optimal f∗ might contain not only OT map T ∗ but other functions as well. However, our experiments (M7) show that this is not a serious issue in practice" only acknowledges this problem, but doesn't explain how it is dealt with.
3) It seems like the final obtained algorithm is not truely different from the standard GAN optimization. There are 3 main differences that are highlighted between the optimization problems, but it seems that in the optimization is the same in terms of the same alternating generator and discriminator loss updating, exept that the 'content' loss has a weight (lambda) of 1.
4) In terms of the results, overall - the method is shown to work, but is in par with other methods including the baseline GANs. The claim that one does not have to pick the lambda parameter isn't very influencial, since the GAN results work well with a particula choice of lambda.
5) In terms of framing, I didn't find anything that was specific to super-resolution (and in fact the method is also demonstrated on a deblurring task). The analysis and solution are relevant to a much wider set of tasks over the unpaired setting of two domains.
6) GAN training suffers from know stability issues. There should be some reference to this aspect of the optimization as well.

**Summary Of The Paper:**

The paper introduces some new ideas and findings, by analyzing GAN-based models that learn constrained mappings between domains, through the perspective of optimal transport. This is important, since such GANs are used extensively for a variety of tasks, e.g. 'inverse' problems like super-resolution, denoising and debluring.

Since the setting is an 'unpaired' one, in addition to the goal of having the 'generated' target distribution close to the 'real' one (this is enforced by the discriminator), there is a need to constrain the generator to keep the output close to the input, which is achieved by a 'cost' (e.g. 'content') function for minimizing such differences.

In this setting, they show that the learned generator (T) is an optimal transport map between the input distribution (e.g. LR images) and the image of T (the generated distribution), under the 'content' cost function. Furthermore, they show that the generator (or transport map) is biased, in the sense that perfect optimization will necessarily not lead to the true target distribution, but rather will be 'biased' towards the input distribution.

As an alternative, they suggest directly optimizing for an optimal transport map between source and target distributions, under the content cost. Several derivations and reasonable assumptions, lead to a formulation and GAN-type optimization scheme, which is verified empirically in several settings.

**Summary Of The Review:**

The paper was very inspiring and interesting to read, especially in terms of the OT point of view on GAN.
However, I am not convinced that the analysis leads to a result that is significant enough in terms of its ability to overcome GAN's weaknesses, especially since the algorithm itself does not seem to differ significanlty from GAN (see above).
The bias of the generator is certainly not the only important factor in GAN quality and the different formulation might effect performance in other ways. The bias claim is empirically condsidered only in Section B of the appendix, only in terms of the color distribution variance, while the qualitative and quantitave results on that data (Aim19 dataset) are not better than those of the other methods.

---

> ### Author Response · Authors · 2022-11-16
> **Answer to Reviewer GsaW**
>
> Thank you for your valuable feedback. Please find above (in our reply to all the Reviewers) the answers to your comments common with other reviews. Please find below our answers to your questions that do not overlap with those of other Reviewers.
>
> **(1) There is a problem with the logic that follows Lemma 3 - It wrongly claims that <...> one can solve a saddle point problem (12), obtain an optimal pair $(f^\star, T^\star)$, and use $T^\star$ as an OT map from $\mathbb{P}$, $\mathbb{Q}$". <...> It was shown that the true $T^\star$ can be chosen as optimal along with the true $f^\star$ , but this doesn't show that the map obtained along with $f^\star$ is actually that one and not another.**
>
> We agree that we were not absolutely accurate in the paragraph following Lemma 3. We corrected it in a revised version, see **Section 6.1**. Please note that we stated the issue you are referring to in the conclusion of our initial submission as a limitation of our approach.
>
> **(2) GAN training suffers from known stability issues. There should be some reference to this aspect of the optimization as well.**
>
> In the revised version of our paper, we include the reference to this aspect in the limitations **(page 9)**.
>
> **(3) The bias claim is empirically considered only in Section B of the appendix, only in terms of the color distribution variance.**
>
> We think there is a misunderstanding. First, we empirically demonstrate the bias in Section 7.1 using the Wasserstein-2 benchmark for which an OT map is analytically known. In this section, the bias is demonstrated not only qualitatively (Figure 5) but also quantitatively using the direct metrics between computed transport map and the true OT map. After that, we additionally demonstrate the bias using color palettes statistics for the solutions of different methods on AIM19 dataset. Importantly, we theoretically prove the existence of a bias in regularized GANs solutions.
>
> **Concluding remarks**. Please respond to our post to let us know if the clarifications above suitably address your concerns about our work. We are happy to address any remaining points during the discussion phase; if the responses above are sufficient, we kindly ask that you consider raising your score.

---

> > ### Author Response · Authors · 2022-12-02
> > **Looking forward to receiving your feedback**
> >
> > Dear Reviewer GsaW,
> >
> > We thank you for your review and greatly appreciate the time you spent reviewing our paper.
> >
> > The end of the rebuttal phase is approaching. We would be grateful if we could hear your feedback regarding our answers to the reviews. We are happy to address any remaining points during the remaining period.
> >
> > Thanks in advance,
> >
> > Paper authors

---

### Official Review · Reviewer_xhtM · 2022-10-24

**Confidence:** 4
**Correctness:** 3
**Technical Novelty And Significance:** 2
**Empirical Novelty And Significance:** 1
**Recommendation:** 3

**Clarity, Quality, Novelty And Reproducibility:**

The writing of the paper is not good and needs a major revision. The authors mix the GANs and SR methods in the writing the paper. I would say that all the theoretical analysis throughout this paper are general and not restrict to the SR problem. However, the authors keep mentioning the LR or SR here and there (Corollary 1 for example). This confuses readers and one would think the analysis is specific to the SR problem.

The description of the experimental results of Table 3 is not clear. In Table 3, there are OTS (ours MSE) and OTS (ours, VGG), but in the text, there is only OTS, I’m not sure which OTS in Table 3 are the authors referring to.

The authors should cite the papers that use Eq. (5) for training GANs.

It is necessary to mention in Theorem 1 that $\epsilon$ is sufficient small enough to make the claim hold.

It is necessary to mention in Algorithm 1 that whether one needs to minimize or maximize the objective, not just mention "using".

Minor:

$\mathcal{D}(\mathbb{Q} + \epsilon \Delta \mathbb{Q})$ should be $\mathcal{F}(\mathbb{Q} + \epsilon \Delta \mathbb{Q})$ in Theorem 1.


**Strength And Weaknesses:**

## Strength:

1. This paper provides details to analyze the transport plan of optimal transport regularized GANs. The observation is that the solution is biased.
2. The connection and difference between the Regularized IPM GANs and the OTM are analyzed.
3. The authors applied the OTM to the image Super Resolution (SR) problem.

## Weakness

1. The theoretical results are not novel.
    1. In Sec. 5, the authors proved that adding a regularization term, the $\mathcal{R}\_c(T)$ in Eq. (5) will not be able to recover the true target distribution $\mathbb{Q}$. This does not seem that novel to me because regularization technique is a well-known technique in machine learning. Adding a regularization term to a loss will prevent the loss term ($\mathcal{D}$) from decreasing to zero, such that the $T^*_{\\#}\mathbb{P}$ does not equal to $\mathbb{Q}$. This is expected. In general, the purpose of the regularization term is to prevent overfitting or stabilize training GANs.

     2. The theoretical results in Sec. 6 are not novel. Almost all results could be found in [1]. If one retain the constants of Eqs. (8) - (11) in [1] and replace $\frac{1}{2} || x - y ||^2$ by $c(x, y)$ and replace $\psi(y)$ by $v(y)$ in [1], one will find that Lemma 3 in this paper is equivalent to Eq. 12 (Lemma 4.1) in [1]. As also mentioned in [1], "we focus on the quadratic ground cost $\frac{1}{2} || x - y ||^2$. Nevertheless, our approach extends to other costs $c(\cdot,\cdot)$". In adddition, Algorithm 1 in this paper is almost the same as Algorithm 1 in [1] by using the identity embedding $Q$, and a general cost $c(\cdot,\cdot)$ in Algorithm 1 of [1].  Unfortunately, the authors did not even mentioned [1] in the whole Sec. 6.

2. The authors made incorrect claim “The solution of the regularized GAN is an OT map” in Lemma 1. The authors tend to mix the optimal transport GAN or Wasserstein GAN as the whole GAN family. However, these GANs are just part of the GAN family. The authors analyzed Biased Optimal Transport in GANs in Sec. 5. However, I want to mention that not all GANs use the optimal transport or the Wasserstein distance, nor do all the GANs uses the optimal transport regularizer as wrote in Eq. (5). Therefore, it is inappropriate to claim that minimizer of a regularized GAN problem is always an OT map as mentioned in the abstract and Lemma 1.

3. The experimental results are not good. On the AIM dataset, the proposed OTS variants do not perform well as the DASR method in Table 3. Also, why not compare with other SR methods such as DASR on the face (CelebA?) dataset?

References:

[1] Rout et al., Generative modeling with optimal transport maps, ICLR 2022.


**Summary Of The Paper:**

This paper analyzed the results of optimal transport regularized GANs, and claimed that the optimal transport map for optimal transport regularized GANs are biased. The authors proposed to use the method in Rout et al. 2022 (OTM) for the image Super Resolution (SR) problem. Experiments are conducted on a face dataset (CelebA?) and the AIM dataset.

**Summary Of The Review:**

This paper lacks novelty and the experimental results are not good.

---

> ### Author Response · Authors · 2022-11-16
> **Answer to Reviewer xhtM**
>
> Thank you for your valuable feedback. Please find above (in our reply to all the Reviewers) the answers to your comments common with other reviews. Please find below our answers to your questions that do not overlap with those of other Reviewers.
>
> **(1) The theoretical results in Sec. 6 are not novel. Almost all results could be found in [1]. <...> Unfortunately, the authors did not even mentioned [1] in the whole Sec. 6.**
>
> We agree with the reviewer that we were not quite accurate in citing the related work [1] in Section 6. However, we think that there is a misunderstanding regarding the novelty of our theoretical results and our algorithm. The algorithm in paper [1] is designed for the (Q-embedded) quadratic cost function and represents a particular case of our algorithm which is suitable for general costs. The authors of [1] indeed mention that their theory might be extended to a general cost function case. However, they do not present the theory and algorithm for a general case and note it as a limitation of their approach. In our paper, we fill this gap both from theoretical and empirical side, see theory (Section 6) and validation on SR task with the perceptual cost function (Section 7).
> To address the reviewer's comment, we mention the connection between our algorithm and that of paper [1] in Section 6 of the revised version of our paper **(page 5)**.
>
> **(2) Not all GANs use the optimal transport or the Wasserstein distance, nor do all the GANs uses the optimal transport regularizer as written in Eq. (5). Therefore, it is inappropriate to claim that minimizer of a regularized GAN problem is always an OT map as mentioned in the abstract and Lemma 1. <...> The authors should cite the papers that use Eq. (5) for training GANs.**
>
> Our Equation (5) represents a basic form of regularized GANs objective. We note that the equation does not specify the type of discrepancy or content loss. Our Lemma 1 is appropriate for GANs **with arbitrary discrepancies $\mathcal{D}$ (not only Wasserstein distance) and with arbitrary content losses $c(x, y)$**.
>
> *As per request, we cite the papers that use Equation (5) for GANs training in **Appendix F** in the revised version of our paper.*
>
> **(3) The description of the experimental results of Table 3 is not clear. In Table 3, there are OTS (ours MSE) and OTS (ours, VGG), but in the text, there is only OTS, I’m not sure which OTS in Table 3 are the authors referring to.**
>
> We clarified the notation in the revised version of our paper, see the **implementation details in Section 7.2**.
>
> **(4) It is necessary to mention in Theorem 1 that $\epsilon$ is sufficient small enough to make the claim hold.**
>
> *We kindly ask the reviewer to specify where, from his point of view, the requirement of the sufficient smallness of $\epsilon$ should be included? From our point of view, the current theorem formulation is correct.*
>
> **(5) It is necessary to mention in Algorithm 1 that whether one needs to minimize or maximize the objective, not just mention "using".**
>
> In the revised version of our paper, we **updated Algorithm 1** by specifying which objective should be minimized and which should be maximized.
>
> **(6) $D(Q+\epsilon\Delta Q)$ should be $\mathcal{F}(Q+\epsilon\Delta Q)$ in Theorem 1.**
>
> We kindly ask the reviewer to detail this comment. From our point of view the designation is correct.
>
> **Concluding remarks**. Please respond to our post to let us know if the clarifications above suitably address your concerns about our work. We are happy to address any remaining points during the discussion phase; if the responses above are sufficient, we kindly ask that you consider raising your score.
>
> **Additional References.**
>
> [1] Litu Rout, Alexander Korotin, and Evgeny Burnaev. Generative modeling with optimal transport maps. In International Conference on Learning Representations, 2022. URL https://openreview.net/forum?id=5JdLZg346L

---

> > ### Comment · Reviewer_xhtM · 2022-11-30
> > **Reply**
> >
> > I would thank the authors for the response! Next I reply to the authors' responses.
> >
> > **"Our paper is a pioneering work showing that all regularized GANs secretly solve the optimal transport problem."**
> >
> > This is not true, or at least not precise. What the authors considered in this paper is a particular regularizer (the primal OT formulation), not a general regularizer. For example, the regularizer in this paper does not include the gradient penalty used in WGAN-GP [2], nor does it include the R1 regularizer in [3]. As I have already pointed out in my initial review, the authors made incorrect claim “The solution of the regularized GAN is an OT map” in Lemma 1.
> >
> > **The algorithm in paper [1] is designed for the (Q-embedded) quadratic cost function and represents a particular case of our algorithm which is suitable for general costs. ... In our paper, we fill this gap both from theoretical and empirical side, see theory (Section 6)**
> >
> > As I have already mentioned in my initial review, Algorithm 1 in this paper is almost the same as Algorithm 1 in [1] by using the identity mapping $Q$, and a general cost in Algorithm 1 of [1]. I would not consider the theoretical results in Sec. 6 and Algorithm 1 in this paper as innovation, because using $Q$ as an identity mapping and using the general cost have already been mentioned in [1], and the derivation is obvious, as I wrote in my initial review.
> >
> > **However, they do not present the theory and algorithm for a general case and note it as a limitation of their approach.**
> >
> >  The authors do not present the theory and algorithm for a general case because they are straightforward. I think the authors misunderstand what does limitation in [1] refer to. Indeed, the authors in [1] mentioned the "our approach extends to other costs" in the "Limitations paragraph" in Sec. 6 of [1]. However, the authors in [1] did not mean the general cost is a limitation, but meant the existence of the OT map between two distributions is a limitation.
> >
> > **Our Lemma 1 is appropriate for GANs with arbitrary discrepancies (not only Wasserstein distance) and with arbitrary content losses.**
> >
> > My concern still remains. Not all GANs uses the optimal transport as the regularizer. For example, the WGAN-GP [2] uses the gradient penalty as the regularizer, and the R1 regularized GAN [3] uses the zero-centered gradient penalty. These cases are not included in the analysis of this paper. Therefore, "The solution of the regularized GAN is an OT map" in Lemma 1 is incorrect.
> >
> > **We kindly ask the reviewer to specify where, from his point of view, the requirement of the sufficient smallness of $\epsilon$ should be included?**
> >
> > In Theorem 1, the $\epsilon$ is sufficiently small should be mentioned, because the authors require $\epsilon$ to be sufficiently small in their proof, quoted as "We see that $\mathcal{F}(\mathbb{Q} + \epsilon \Delta \mathbb{Q})$ is smaller then $\mathcal{F}(\mathbb{Q})$ for sufficiently small $\epsilon > 0$".
> >
> > **For $\mathcal{D}(\mathbb{Q} + \epsilon \Delta \mathbb{Q})$ should be $\mathcal{F}(\mathbb{Q} + \epsilon \Delta \mathbb{Q})$ in Theorem 1, we kindly ask the reviewer to detail this comment. From our point of view the designation is correct.**
> >
> > In theorem 1, the authors wrote "$\mathcal{D}(\mathbb{Q} + \epsilon \Delta \mathbb{Q}) = \mathcal{D}(\mathbb{Q}, \mathbb{Q}) + o(\epsilon)$". I think $\mathcal{D}(\mathbb{Q} + \epsilon \Delta \mathbb{Q})$ should be $\mathcal{F}(\mathbb{Q} + \epsilon \Delta \mathbb{Q})$. According to definition of $\mathcal{D}$ in one line above Eq. (5). There, $\mathcal{D}$ takes two arguments as inputs, measuring the discrepancy between two distributions. Here, $\mathcal{D}(\mathbb{Q} + \epsilon \Delta \mathbb{Q})$ takes only one argument as input. I do not understand the meaning of it. If the authors think $\mathcal{D}(\mathbb{Q} + \epsilon \Delta \mathbb{Q})$ is correct. I would kindly ask the authors to write the definition of $\mathcal{D}$ that takes only one argument as input and the meaning of it.
> >
> > **According to the correctly highlighted quantitative results in Table 3 there is no universal method that beats every other according to all metrics.**
> >
> > I understand that no method wins at all metrics, but I would say losing 6 (OTS (MSE) under all four metrics and OTS (VGG) under PSNR, LPIPS) out of 8 (OTS (MSE) and OTS (VGG) under all four metrics) compared to DASR in Table 3 are weak experimental results.
> >
> > **we cite the papers that use Equation (5) for GANs training in Appendix F in the revised version of our paper.**
> >
> > None of the papers in Appendix F use Eq. (5) for image super-resolution. Remember that the regularizer in Eq. (5) is primal OT. No paper in Appendix F uses the OT regularizer.
> >
> > ### References:
> >
> > [1] Rout et al., Generative modeling with optimal transport maps, ICLR 2022.
> >
> > [2] Gulrajani et al., Improved Training of Wasserstein GANs, NeurIPS 2017.
> >
> > [3] Mescheder et al., Which Training Methods for GANs do actually Converge?, ICML 2018.

---

> > > ### Author Response · Authors · 2022-12-02
> > > **Answer to Reviewer xhtM (1/2)**
> > >
> > > Thank you for the detailed analysis of our paper and feedback. We greatly appreciate the time you spent reviewing our paper. However, we think there might be a misunderstanding of our paper results and contributions. Please find below our clarifications.
> > >
> > > **(1) What the authors considered in this paper is a particular regularizer (the primal OT formulation), not a general regularizer. <...> Not all GANs uses the optimal transport as the regularizer. For example, the WGAN-GP [2] uses the gradient penalty as the regularizer, and the R1 regularized GAN [3] uses the zero-centered gradient penalty. These cases are not included in the analysis of this paper. Therefore, "The solution of the regularized GAN is an OT map" in Lemma 1 is incorrect.**
> > >
> > > Our paper *only considers GANs with content loss as the regularization for the generator*. This is clearly specified in our contributions (Section 1, Section 8) and Section 5 of our paper. The analysis of this type of regularized GANs is important because they are widely used in unpaired SR task (Section 2, Appendix F). Both papers [2], [3] that you mentioned consider *regularizations for the discriminator*, i.e., gradient penalty and $R_1$ regularizations in order to enforce 1-Lipschitz constraint on the discriminator and to improve the stability of GAN training, respectively.
> > >
> > > *The issues raised in papers [2], [3] and regularizations that you mentioned are not relevant to our study.*
> > >
> > > **(2) Algorithm 1 in this paper is almost the same as Algorithm 1 in [1] by using the identity mapping $\mathbb{Q}$, and a general cost in Algorithm 1 of [1]. I would not consider the theoretical results in Sec. 6 and Algorithm 1 in this paper as innovation, because using $\mathbb{Q}$
> > > as an identity mapping and using the general cost have already been mentioned in [1], and the derivation is obvious, as I wrote in my initial review. <...> The authors in [1] mentioned the "our approach extends to other costs" in the "Limitations paragraph" in Sec. 6 of [1]. However, the authors in [1] did not mean the general cost is a limitation, but meant the existence of the OT map between two distributions is a limitation.**
> > >
> > > If the reviewer considers the part with derivation of the algorithm to be simple and easily following from the prior work, we can soften our claims related to this part of our paper. Still, we think that the rest of the paper presents *novel and significant* analysis of the relation between dual OT methods and GANs with "primal OT" (content loss) regularizer.
> > >
> > > **(3) In Theorem 1, the $\epsilon$ is sufficiently small should be mentioned, because the authors require $\epsilon$ to be sufficiently small in their proof, quoted as "We see that $\mathcal{F}(\mathbb{Q}+\epsilon\Delta\mathbb{Q})$ is smaller then $\mathcal{F}(\mathbb{Q})$ for sufficiently small $\epsilon>0$.**
> > >
> > > The sufficient smallness of $\epsilon$ is not a requirement of our Theorem 1. Indeed, $\epsilon\geq0$ in the formulation of the theorem (definition of the first variation) must satisfy only one condition, i.e., $\mathbb{Q}+\epsilon\Delta\mathbb{Q}\in\mathcal{P(Y)}$. In the proof of the theorem, we consider particular $\Delta \mathbb{Q}=\mathbb{P}-\mathbb{Q}$ for which $\forall\epsilon\in[0,1]$ the condition $\mathbb{Q}+\epsilon\Delta\mathbb{Q}\in\mathcal{P(Y)}$ is automatically satisfied. We then tend $\epsilon$ to zero to show that among all $\epsilon\in[0,1]$ there are ones, such that $\mathcal{F}(\mathbb{Q}+\epsilon\Delta\mathbb{Q})$ is smaller then $\mathcal{F}(\mathbb{Q})$. As can be seen from the proof, this is true for all $\epsilon$ in some neighborhood of zero, i.e., for all sufficiently small $\epsilon$.
> > >
> > > **(4) In theorem 1, the authors wrote "$\mathcal{D}(\mathbb{Q}+\varepsilon \Delta \mathbb{Q})=\mathcal{D}(\mathbb{Q}, \mathbb{Q}) + o(\varepsilon)$". I think $\mathcal{D}(\mathbb{Q}+\varepsilon \Delta \mathbb{Q})$ should be $\mathcal{F}(\mathbb{Q}+\varepsilon \Delta \mathbb{Q})$. According to definition of $\mathcal{D}$ in one line above Eq. (5). There, $\mathcal{D}$ takes two arguments as inputs, measuring the discrepancy between two distributions. Here, $\mathcal{D}(\mathbb{Q}+\varepsilon \Delta \mathbb{Q})$ takes only one argument as input. I do not understand the meaning of it.**
> > >
> > > We thank the reviewer for pointing out the typo. We accidentally did not write down the second argument of a discrepancy $\mathcal{D}(\mathbb{Q}+\varepsilon \Delta \mathbb{Q}, \mathbb{Q})=\mathcal{D}(\mathbb{Q}, \mathbb{Q}) + o(\varepsilon)$. We will fix this typo in the final version of our paper.

---

> > > > ### Author Response · Authors · 2022-12-02
> > > > **Answer to Reviewer xhtM (2/2)**
> > > >
> > > > **(5) I understand that no method wins at all metrics, but I would say losing 6 (OTS (MSE) under all four metrics and OTS (VGG) under PSNR, LPIPS) out of 8 (OTS (MSE) and OTS (VGG) under all four metrics) compared to DASR in Table 3 are weak experimental results.**
> > > >
> > > > Our main experimental result is OTS with the **perceptual** cost, i.e. OTS (VGG). It beats DASR according to SSIM, and, importantly, FID which far better correlates with perceptual quality as we mentioned in Section 7.1, page 7. We included in comparison OTS with MSE cost just to show the advantages of using the **perceptual** cost function.
> > > >
> > > > In view of your above comment (2), we would like to emphasize that *the key contribution of our paper is the theoretical analysis* in Section 5. The main purpose of our experiments is to empirically demonstrate theoretically revealed findings about the bias issue.
> > > >
> > > > **(6) None of the papers in Appendix F use Eq. (5) for image super-resolution. Remember that the regularizer in Eq. (5) is primal OT. No paper in Appendix F uses the OT regularizer.**
> > > >
> > > > If we understand correctly, by the phrase "OT regularizer" you mean exactly the content loss regularizer from Equation (5). Our paper analyses generic learning objectives of the form "Loss = **GAN loss + content loss** (OT regularizer)". Thus, we do not understand your concern as every paper in Appendix F optimizes an objective of this kind.
> > > >
> > > > **Additional References.**
> > > >
> > > > [1] Rout et al., Generative modeling with optimal transport maps, ICLR 2022.
> > > >
> > > > [2] Gulrajani et al., Improved Training of Wasserstein GANs, NeurIPS 2017.
> > > >
> > > > [3] Mescheder et al., Which Training Methods for GANs do actually Converge?, ICML 2018.

---

### Author Response · Authors · 2022-11-16
**General response**

Dear reviewers,

We thank you for your thoughtful comments! We are glad that you positively highlight our theoretical insights (Reviewer GsaW, nWwQ), empirical analysis approving our theoretical conclusions (Reviewer nWwQ), clear structure (Reviewer GsaW, mWTP, nWwQ) and understandable derivations (Reviewer GsaW, nWwQ). We are inspired by your comment that our approach shows comparable to state-of-the-art results on the AIM19 dataset (Reviewer nWwQ). We hope that our developed Optimal Transport Solver **(OTS)** approach for unpaired image super-resolution problem will be convenient to setup and use in practical applications.
Please, find the answers to your shared questions below.

**(1) Performance of the proposed method on AIM19 dataset. (Reviewer xhtM, GsaW)**

We accidentally highlighted the 1st, 2nd and 3d best results in Table 3 incorrectly in the initial submission. We corrected this in the revised version. We kindly ask the reviewers to reconsider the **updated table**.
 According to the correctly highlighted **quantitative** results in Table 3 there is **no universal method** that beats every other according to all metrics. Among FSSR and DASR methods, FSSR achieves better FID, while DASR is superior according to LPIPS. Our method with perceptual transport cost beats FSSR according to LPIPS, and DASR according to FID. Thus, OTS method performs comparably to FSSR and DASR while it is one-step and has much less hyper-parameters, see Table 5.

We also kindly suggest the reviewers to reconsider the **qualitative** results of our method from a computer screen. FSSR and DASR methods results in Figure 6 contain artifacts, which are well visible in lines 3, 4 of the Figure. Moreover, results of FSSR (and DASR) methods in Figure 6 are far less contrast than the results of our OTS method and HR images. This insight is further approved by the results in Figure 7. We note that the presented image crops positions are common for papers on image SR, see FSSR (Fritsche et al., 2019), DASR (Wei et al., 2020), (Yuan et al., 2018), (Kim et al., 2020).

**(2) The theoretical analysis throughout this paper is general and not restricts to the SR problem. (Reviewer xhtM, GsaW)**

As we have pointed in the conclusion of our paper, the presented theoretical analysis is indeed general, and the proposed method could be applied to other unpaired tasks as well.
However, in our work we focus on the unpaired SR problem because for this problem the
presence of a content loss is intuitively understandable.
Addressing your comment, we added the additional note about the usage of SR notation in the main text of our paper **(page 3)**.

**(3) Bias due to the regularization technique is not a novel observation. (Reviewer xhtM, GsaW)**

We agree that bias in the solutions of the regularized ML methods is quite an intuitive observation. But we are not aware of any paper studying it for regularized GANs.
While for simple methods like linear regression the solutions of its regularized versions are explicitly known, the same is not true for regularized GANs. Most of the papers on regularized GANs are purely practical and do not assess what mapping is actually being learned. Our paper is a pioneering work showing that **all regularized GANs secretly solve the optimal transport problem**. This observation provides a new point of view on the problems approached by regularized GANs.

*To conclude, our study shows the importance of the future development of scalable OT methods as anyway regularized GANs try to solve OT problem.*

**(4) Difference between OT optimization algorithm and that of regularized GANs. (Reviewer GsaW, mWTP)**

Technically, we solve a different problem to that of the GANs. We use stochastic gradient ascent-descent algorithm as in GANs because it is a standard approach to handle saddle point problems (Chambolle et al., 2016). The crucial difference with GANs is that in our algorithm the discriminator (potential) is in the outer optimization cycle and generator (transport map) $-$ in the inner. It means, that for 1 step of the discriminator the generator weights are updated multiple times. The situation is absolutely opposite for GANs. Thus, while being similar to GANs optimization algorithm, our algorithm solves inherently different optimization problem.

---

> ### Author Response · Authors · 2022-11-16
> **General response. Additional references**
>
> Antonin Chambolle and Thomas Pock. An introduction to continuous optimization for imaging. Acta Numerica, 25, pp. 161-319, 2016.
>
> Gwantae Kim, Jaihyun Park, Kanghyu Lee, Junyeop Lee, Jeongki Min, Bokyeung Lee, David K. Han, and Hanseok Ko. Unsupervised real-world super resolution with cycle generative adversarial network and domain discriminator. In Proceedings of the IEEE/CVF Conference on Computer Vision and Pattern Recognition Workshops (CVPRW), pp. 1862–1871, June 2020.
>
> Wei Wang, Haochen Zhang, Zehuan Yuan, and Changhu Wang. Unsupervised real-world super- resolution: A domain adaptation perspective. In Proceedings of the IEEE/CVF International Conference on Computer Vision, pp. 4318–4327, 2021.
>
> Yuan Yuan, Siyuan Liu, Jiawei Zhang, Yong bing Zhang, Chao Dong, and Liang Lin. Unsupervised image super-resolution using cycle-in-cycle generative adversarial networks. 2018 IEEE/CVF Conference on Computer Vision and Pattern Recognition Workshops (CVPRW), pp. 814–81409, 2018.

---

> > ### Author Response · Authors · 2022-11-16
> > **Revision of the paper**
> >
> > We have uploaded an updated version of the paper. The newly added content is highlighted with the **blue** color. The changes are:
> >
> > - **[xhtM, GsaW]** Fix in the highlight of **Table 3**
> >
> > - **[xhtM, GsaW]** Additional note about the usage of SR notation, see **Section 5 (page 3)**
> >
> > - **[xhtM]** New **Appendix F** showing the connection between equation (5) and learning objectives of popular SR methods; Added reference to Appendix F, see **Section 5 (page 3)**
> >
> > - **[xhtM]** Added reference to the paper by (Rout et al., 2022), see **Section 6 (page 5)**
> >
> > - **[xhtM]** New details in Algorithm 1 showing whether the objectives should be minimized or maximized **(page 7)**
> >
> > - **[xhtM]** Change in OTS method designation, see **Section 7.2 (pages 8-9, Table 3, Figures 6)** and **Appendix E (Figures 9-10)**
> >
> > - **[GsaW]** Added note about possible limitations of OTS similar to GANs, see **Section 8 (page 10)**.

---

### Author Response · Authors · 2022-12-09
**Concluding remarks**

Dear Area Chair and Reviewers,

Before the discussion closes, we want to summarize the discussion and evidence showing the novelty and significance of our work.

Existing papers on GANs regularized with content losses are mostly practical and do not assess what kind of mapping they are actually learning. Probably, this analysis have not been done due to its complexity on the space of probability distributions. We present a *novel optimal transport point of view* on GANs regularized with content losses which are used in many **practically important** computer vision problems, e.g., super-resolution. We theoretically prove that these GANs actually learn an OT mapping which, however, is usually biased (Section 5). We justified our analysis by experiments with the Wasserstein-2 benchmark (Section 7.1) and the large-scale dataset for unpaired SR (Section 7.2).

Overall, our analysis emphasizes the **importance of developing OT methods** since GANs regularized with content losses anyway try to solve OT problem. We note that such a new way of thinking about various deep generative models is gaining popularity. For example, a recent paper entitled ["Understanding DDPM Latent Codes Through Optimal Transport"](https://openreview.net/forum?id=6PIrhAx1j4i) which was submitted to ICLR 2023 analyzes *diffusion models from an optimal transport perspective*. This analysis shows that DDPM encoder map coincides with the optimal transport map for common distributions.

Moreover, while OT optimization objectives (such as our Eq. (12)) are getting increasing attention from the ML community (Rout et al., 2021), (Fan et al., 2022), (Korotin et al., 2021, 2022), their connection with optimization objectives of GANs is not yet explored. Our paper establishes this connection stating both similarities and differences between OT methods and GANs regularized with content losses and their underlying learning principles (Section 6.2, Table 1).

We kindly ask you to take these important aspects into account when making the final decision.

**Additional References**

Rout, L., Korotin, A., \& Burnaev, E. (2021, September). Generative Modeling with Optimal Transport Maps. In International Conference on Learning Representations.

Fan, J., Liu, S., Ma, S., Zhou, H. \& Chen, Y. (2022). Neural Monge Map estimation and its applications. arXiv preprint arXiv:2106.03812.

Korotin, A., Li, L., Genevay, A., Solomon, J., Filippov, A., \& Burnaev, E. (2021, May). Do Neural Optimal Transport Solvers Work? A Continuous Wasserstein-2 Benchmark. In Advances in Neural Information Processing Systems.

Korotin, A., Kolesov, A., \& Burnaev, E. (2022). Kantorovich Strikes Back! Wasserstein GANs are not Optimal Transport?. In Thirty-sixth Conference on Neural Information Processing Systems Datasets and Benchmarks Track.

---

### Decision · Program_Chairs · 2023-01-20

**Decision:**

Reject

**Justification For Why Not Higher Score:**

* the proof of theorem overlooks the role of o(\epsilon)
* the choice of the regularizer is not clearly and carefully discussed. why a Monge map cost regularizer is chosen?
* experimental results are a bit weak compared to state of the art

**Justification For Why Not Lower Score:**

N/A

**Metareview: Summary, Strengths And Weaknesses:**

The paper proposes a OT point of view of image superresolution. This new perspective comes naturally from the choice
of the content loss regularizer of the authors.


The paper has been discussed (live) and it has been concluded that several concerns still need to
be addressed before acceptance, despite some positive points. Weaknesses that have mentioned are :

* The choice of the content loss is not clearly justified while being critical for the contribution.
most approaches in the literature use losses at the pixel level (eg l2 loss). We expect a better justification
for the choice in Equation 1, as this choice corresponds to a OT loss and makes the rest of the analysis
and derivation natural (I mean : it is natural that regularized gan is an OT problem if the regularization
is an Monge map loss)
* Experiments are not fully convincing. In Table 2, one would have expected comparisons with some other
superresolution algorithm. Performance in the large-scale evalution are far from being compelling.
* From a theoretical point of view, the impact of o(\epsilon) in the proof of Theorem 1 needs to
be analysed better. How sufficiently small it is and do we have guarantee about that?

**Summary Of Ac-Reviewer Meeting:**

All the above three points have been raised and addressed during the meeting. The first and last points have not been clarified by the authors and are critical for the paper